# Multi-Domain Long-Tailed Learning by Augmenting Disentangled Representations

**Xinyu Yang**[*]                                                  *xinyuya2@andrew.cmu.com*
*Carnegie Mellon University*

**Huaxiu Yao**[*]                                                  *huaxiu@cs.unc.edu*
*University of North Carolina at Chapel Hill*

**Allan Zhou**                                                  *ayz@cs.stanford.edu*
*Stanford University*

**Chelsea Finn**                                                  *cbfinn@cs.stanford.edu*
*Stanford University*

**Reviewed on OpenReview:** *https://openreview.net/forum?id=4UXJhNSbwd*

## Abstract

There is an inescapable long-tailed class-imbalance issue in many real-world classification problems. Current methods for addressing this problem only consider scenarios where all examples come from the same distribution. However, in many cases, there are multiple domains with distinct class imbalance. We study this multi-domain long-tailed learning problem and aim to produce a model that generalizes well across all classes and domains. Towards that goal, we introduce TALLY, a method that addresses this multi-domain long-tailed learning problem. Built upon a proposed selective balanced sampling strategy, TALLY achieves this by mixing the semantic representation of one example with the domain-associated nuisances of another, producing a new representation for use as data augmentation. To improve the disentanglement of semantic representations, TALLY further utilizes a domain-invariant class prototype that averages out domain-specific effects. We evaluate TALLY on several benchmarks and real-world datasets and find that it consistently outperforms other state-of-the-art methods in both subpopulation and domain shift.

## 1 Introduction

Deep classification models can struggle when the number of examples per class varies dramatically (Beery et al., 2020; Zhang et al., 2021). This *long-tailed* setting arises frequently in practice, such as wildlife recognition (Beery et al., 2020). Classifiers tend to be biased towards majority classes and perform poorly on class-balanced test distributions, i.e. when there is a shift in the label distribution between training and test. Existing approaches address the long-tailed problem by modifying the data sampling strategy (Chawla et al., 2002; Zhang & Pfister, 2021), adjusting the loss function for different classses (Cao et al., 2019; Hong et al., 2021), or augmenting minority classes (Chou et al., 2020; Zhong et al., 2021).

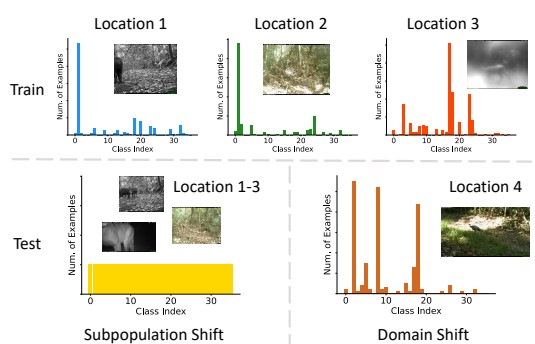

Figure 1: Illustration of imbalanced class distributions across domains in iWildCam, a wildlife recognition benchmark (Beery et al., 2020).

---

[*]Equal contribution. This work was done when Xinyu Yang was remotely mentored by Huaxiu Yao.

Unlike these works, which focus on single-domain long-tailed learning, we study *multi-domain long-tailed learning*, where each domain has its own long-tailed distribution. For example, in wildlife recognition (Figure 1), images are collected from various locations and the distribution of species at each location is typically imbalanced and also varies between locations.

In multi-domain long-tailed classification, classifiers need to handle distribution shift amidst class imbalance. We focus on two types of shifts: subpopulation shift and domain shift. In subpopulation shift, we train a model on data from multiple domains and evaluate the model on a test set with balanced domain-class pairs. For example, in wildlife recognition, a species may be concentrated at only a few locations, creating a spurious correlation between the label (species) and the domain (location). A model trained on the entire population may fail on the test set when this correlation does not hold anymore. In domain shift, we expect the trained model to generalize well to completely new test domains. For example, in wildlife recognition, we train a model on data from a fixed set of training locations and then deploy the model to new test locations.

Prior long-tailed classification methods work well in single-domain settings, but may perform poorly when the test data is from underrepresented domains or novel domains. Meanwhile, invariant learning approaches alleviate cross-domain performance gaps by learning representations or predictors that are invariant across different domains (Arjovsky et al., 2019; Li et al., 2018). Yet, these approaches are mostly evaluated in class-balanced settings, where models must be trained on plenty of examples from each class even if augmentation strategies are applied (Yao et al., 2022) – see a detailed discussion in Appendix B. With multi-domain long-tailed data, learning a class-unbiased domain-invariant model is not trivial since the imbalance can exist within a domain or across domains. We aim to address these challenges in this work by proposing **TALLY** (mul**T**i-dom**A**in **L**ong-tailed learning with ba**L**anced representation reassembl**Y**).

TALLY empowers augmentation to balance examples over domains and classes by decomposing and reassembling example pairs, combining the class-relevant semantic information of one example with the domain-associated nuisances of another Zhou et al. (2022). Specifically, TALLY first decouples the representation of each example into semantic information and nuisances with instance normalization. To further mitigate the effects of nuisances, we first average out domain information over examples of the same class and construct class prototype representations. Each semantic representation is then linearly interpolated with a corresponding class prototype. The domain-associated factors are similarly interpolated with class-agnostic domain factors to improve training stability and remove noise. Finally, TALLY produces augmented representations by reassembling the prototype-enhanced semantic representation and domain-associated nuisances among examples. To further achieve balanced augmentation, we propose a selective balanced sampling strategy to draw example pairs for augmentation. In this way, TALLY encourages the model to learn a class-unbiased invariant predictor.

In summary, our major contributions are: we investigate an important yet less explored problem - multi-domain long-tailed learning, and propose an effective augmentation algorithm called TALLY to simultaneously address the class-imbalance issue and learn domain-invariant predictors. Our approach, TALLY, outperforms existing single-domain long-tailed learning and domain-invariant learning approaches, with a significant error decrease of 5.18% over all datasets. Additionally, TALLY is able to capture stronger invariant predictors compared to prior invariant learning approaches.

## 2    Formulations and Preliminaries

**Long-Tailed Learning.** In this paper, we investigate the setting where one predicts the class label $y \in \mathcal{C}$ based on the input feature $x \in \mathcal{X}$, where $\mathcal{C} = \{1, \ldots, C\}$. Given a machine learning model $f$ parameterized by parameter $\theta$ and a loss function $\ell$, empirical risk minimization (ERM) trains such a model by minimizing average loss over all training examples as

$$\min_{\theta} \mathbb{E}_{(x,y) \sim P^{tr}}[\ell(f_\theta(x), y)], \tag{1}$$

which works well when the label distribution is approximately uniform. In long-tailed learning, however, the label distribution is long-tailed, where a small proportion of classes have massive labels and the rest of classes are associated with a few examples. Assume $\{(x_i, y_i)\}_{i=1}^{N}$ is a training set sampled from training

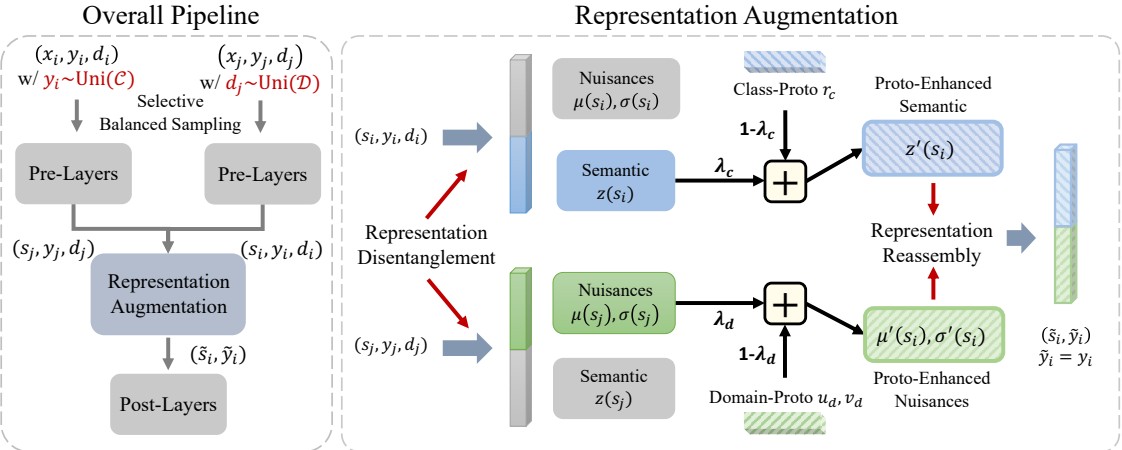

Figure 2: An illustration of TALLY. **Left**: the overall approach produces augmented representations from a pair of examples $x_i$ and $x_j$. At a chosen layer, it mixes their semantic and nuisance information to create augmented representation $\tilde{s}$. **Right**: In detail, the augmentation step disentangles hidden representations $s_i$ and $s_j$ into separate semantic and nuisance factors. It interpolates these with domain-invariant or class-invariant prototypes (respectively) for more robust disentanglement. Finally, it combines the semantic information from $s_i$ with the nuisance information from $s_j$ to create $\tilde{s}_i$.

distribution and the number of examples for each class is $\{n_1, \ldots, n_C\}$, where $\sum_{c=1}^{C} n_c = N$. In long-tailed learning, all classes are sorted according to cardinality (i.e., $n_1 \ll n_C$) and the imbalance ratio $\rho$ is defined as $\rho = n_C/n_1 > 1$. Note that same definitions are used in the test set $\{(x_i, y_i)\}_{i=1}^{N^{ts}}$. Under the class-imbalanced training distribution, vanilla ERM model tends to perform poorly on minority classes, but we expect the model can perform consistently well on all classes.

**Multi-Domain Imbalanced Learning.** Multi-domain long-tailed learning is a natural extension of classical long-tailed learning, where the overall data distribution is drawn from a set of domains $\mathcal{D} = \{1, \ldots, D\}$ and each domain $d$ is associated with a class-imbalanced dataset $\{(x_i, y_i, d)\}_{i=1}^{N_d}$ drawn from domain-specific distribution $p_d$. Following (Albuquerque et al., 2019; Koh et al., 2021), both training and test distribution can be formulated as a mixture distribution over domain space $\mathcal{D}$, i.e., $P^{tr} = \sum_{d=1}^{D} \eta_d^{tr} P_d^{tr}$ and $P^{ts} = \sum_{d=1}^{D} \eta_d^{ts} P_d^{ts}$. The corresponding training and test domains are $\mathcal{D}^{tr} = \{d \in \mathcal{D} | \eta_d^{tr} > 0\}$ and $\mathcal{D}^{ts} = \{d \in \mathcal{D} | \eta_d^{ts} > 0\}$, respectively, where $\eta_d^{tr}$ and $\eta_d^{ts}$ represent the mixture probability. For each domain $d$, we define the number of training examples in each class as $\{n_{1,d}, \ldots, n_{C,d}\}$, sorted by cardinality. The imbalance ratio $\rho^{tr}$ is extended to domain-level ratio as $\rho_d^{tr} = n_{C,d}/n_{1,d}$. During test time, we consider two kinds of test distributions, corresponding to two categories of distribution shifts – subpopulation shift and domain shift. In subpopulation shift, the test domains have been observed during training time, but the test distribution is class-balanced and domain-balanced, i.e., $\mathcal{D}^{ts} \subseteq \mathcal{D}^{tr}$ and $\{\eta_d^{ts} = 1/|\mathcal{D}^{ts}| | \forall d \in \mathcal{D}^{ts}\}$. In domain shift, the test domains are disjoint from the training domains, i.e., $\mathcal{D}^{tr} \cap \mathcal{D}^{ts} = \emptyset$.

## 3 Multi-Domain Long-Tailed Learning with Balanced Representation Reassembly

To improve robustness in multi-domain long-tailed learning, we propose TALLY, a method that can learn class-unbiased, domain-invariant representations through balanced augmentation over classes and domains. The key idea behind TALLY is that every example can be decomposed into *class-relevant semantic* information and domain-associated *nuisances* that should be ignored by an ideal classifier. Nuisances are defined as "class-agnostic" transformations that apply similarly to all classes, such as image style and background changes in image classification Zhou et al. (2022) As outlined in Figure 2, TALLY assumes that domain-associated nuisance information can be transferred among examples. It leverages this to perform augmentation by transferring domain-specific nuisance factors between classes with a novel selective balanced sampling strategy. In the following sections, we will explain in detail how TALLY achieves balanced representation augmentation.

### 3.1 Representation Disentanglement and Reassembly

As described above, TALLY reassembles augmented examples from pairs of examples by combining the semantic representation of one with the domain-related nuisance factors of the other. Motivated by style transfer (Huang & Belongie, 2017), we use instance normalization (InstanceNorm (Ulyanov et al., 2016)) to perform the required disentanglement of semantic and nuisance information. Concretely, given an example $(x, y, d)$ we denote the hidden representation at layer $r$ as $s = f^r(x) \in \mathbb{R}^{C \times H \times W}$, where $C$, $H$, and $W$ denote channel, height, and width dimensions, respectively. InstanceNorm normalizes the example as:

$$z(s) = \text{InstanceNorm}(s) = \frac{s - \mu(s)}{\sigma(s)}, \tag{2}$$
$$\text{where } z(s), \mu(s), \sigma(s) \in \mathbb{R}^C$$

where $\mu(\cdot), \sigma(\cdot)$ are the mean and standard deviation computed across the spatial dimensions of $s$:

$$\mu(s) = \frac{1}{HW} \sum_{h=1}^{H} \sum_{w=1}^{W} s[:, h, w],$$
$$\sigma(s) = \sqrt{\frac{1}{HW} \sum_{h=1}^{H} \sum_{w=1}^{W} (s[:, h, w] - \mu(s))^2}. \tag{3}$$

Following Huang & Belongie (2017), we treat the normalized example $z(s)$ as the semantic representation, and regard $\mu(s)$ and $\sigma(s)$ as the domain-associated nuisances. Notice that we adopt a warm start strategy of running vanilla ERM for the first few epochs to ensure reliable disentanglement.

After decoupling representations, we produce an augmented representation from a pair of examples $(x_i, y_i, d_i)$ and $(x_j, y_j, d_j)$ by swapping semantic representations and domain-associated nuisances:

$$\tilde{s} = \sigma(s_j) \left( \frac{s_i - \mu(s_i)}{\sigma(s_i)} \right) + \mu(s_j), \ \tilde{y} = y_i. \tag{4}$$

Since the semantic content of the augmented representation $\tilde{s}$ is from example $(x_i, y_i, d_i)$, we label our augmented example with $\tilde{y} = y_i$. By reassembling disentangled representations, we can augment representations for minority domains or minority classes.

### 3.2 Selective Balanced Sampling

In the process of representation disentanglement and reassembly, finding a suitable strategy of sampling examples from the training distribution is crucial to solving the domain-class imbalance problem. In single-domain long-tailed learning, up-sampling examples from minority classes is a classical yet effective method. In multi-domain long-tailed learning, the straightforward extension is up-sampling examples from minority domain-class groups, which is named balanced sampling. In practice, for each example $(x_i, y_i, d_i)$, the label $y_i$ and domain $d_i$ are uniformly sampled a joint uniform distribution over all domain-class combinations, i.e., $(y_i, d_i) \sim \text{Uniform}(\mathcal{C}, \mathcal{D})$.

However, to transfer knowledge between different domain-class groups in TALLY, using such a sampling strategy may overemphasize the importance of minority domain-class groups. In Figure 3, we illustrate three domain-class groups from OfficeHome-LT, which is a long-tailed variant of the OfficeHome dataset (Venkateswara et al., 2017). To augment minority groups (e.g., fan-clipart pair), balanced sampling tends to repeatedly draw examples from the same minority group. This is not desirable because it limits the sample diversity in knowledge transfer, and as shown in Figure 3, minority groups typically perform worse than majority groups, making the knowledge transfer less reliable. Therefore, we propose a selective balanced sampling strategy in TALLY. Specifically, for a pair of examples $(x_i, y_i, d_i)$ and $(x_j, y_j, d_j)$, the label $y_i$ of example $i$ is uniformly sampled from all classes ($y_i \sim \text{Uniform}(\mathcal{C})$) and the domain $d_j$ of example $j$ is uniformly sampled from all domains ($d_j \sim \text{Uniform}(\mathcal{D})$). Figure 3 shows that selective balanced sampling has a higher chance of diversifying the sample selection in transferring domain and class information.

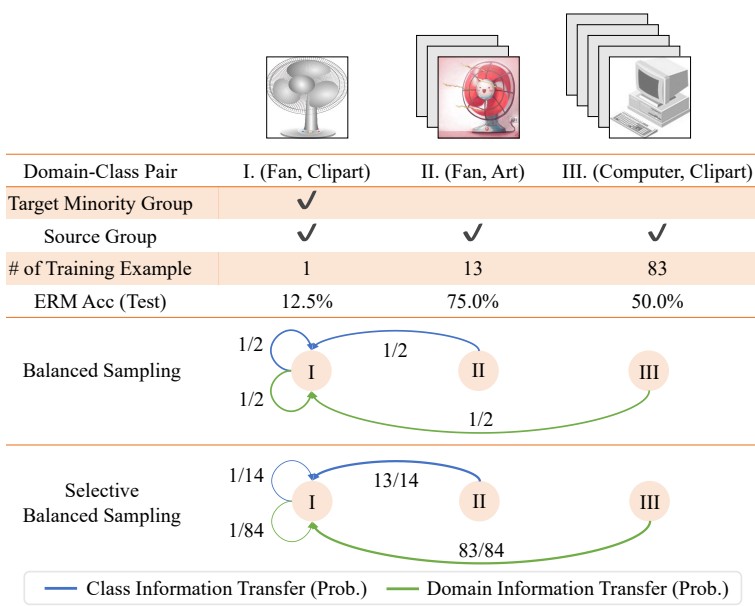

| Domain-Class Pair | I. (Fan, Clipart) | II. (Fan, Art) | III. (Computer, Clipart) |
|---|---|---|---|
| Target Minority Group | ✓ | | |
| Source Group | ✓ | ✓ | ✓ |
| # of Training Example | 1 | 13 | 83 |
| ERM Acc (Test) | 12.5% | 75.0% | 50.0% |

Figure 3: Illustration of selective balanced sampling. Transition probabilities between different pairs are visualized. Detailed discussion is in Appendix A.1.

---

**Algorithm 1** TALLY Training Process

---

**Require:** learning rates $\eta$; warm start epochs $T_0$; prototype momentum $\gamma$; model $f_\theta(\cdot)$ with hidden representation $f_\theta^r(\cdot)$ at layer $r$; Dataset $\mathcal{D}^{tr} = \{(x, y, d)\}$

1: Initialize domain-agnostic prototypes $\{r_c^{(0)}\}_{c=1}^C$ and class-agnostic statistics $\{(u_d^{(0)}, v_d^{(0)})\}_{d=1}^D$
2: Train $f_\theta$ with ERM for $t < T_0$
3: **for** $t = T_0$ to $T$ **do**
4:      $y_i \sim \text{Uniform}(\mathcal{C})$, $d_j \sim \text{Uniform}(\mathcal{D})$
5:      Sample $(x_i, y_i, d_i) \sim \{\mathcal{D}^{tr}|y = y_i\}$, $(x_j, y_j, d_j) \sim \{\mathcal{D}^{tr}|d = d_j\}$
6:      Compute hidden representations $s_i$ and $s_j$
7:      Disentangle semantic factor and obtained $z(s_i)$
8:      Obtaine Enhanced semantic factor $z'(s_i)$ by Eqn. (6)
9:      Enhance nuisance by Eqn. (8) and obtained $(\mu'(s_j), \sigma'(s_j))$
10:     Generate augmented example $(\tilde{s}', \tilde{y}')$ by Eqn. (9)
11:     Train the model on augmented example by Eqn. (10)
12:     Estimate the current prototypes and feature statistics $\{r_c\}_{c=1}^C$, $\{(u_d, v_d)\}_{d=1}^D$
13:     **for** $c = 1$ to $C$ **do**
14:        $r_c^{(t+1)} \leftarrow \gamma r_c^{(t)} + (1 - \gamma)r_c$
15:     **end for**
16:     **for** $d = 1$ to $D$ **do**
17:        $(u_d^{(t+1)}, v_d^{(t+1)}) \leftarrow \gamma(u_d^{(t)}, v_d^{(t)}) + (1 - \gamma)(u_d, v_d)$
18:     **end for**
19: **end for**

---

### 3.3 Prototype-guided Invariant Learning

Since the semantic representation $z(s)$ (Eqn. (2)) should contain only class-relevant information, it should ideally be *domain-invariant*. However, per-instance statistics can be noisy and instance normalization may not perfectly disentangle the semantic information from the domain-related nuisances. To improve robustness, we can "average out" domain information over many examples of the same class from different domains. However, merely averaging over examples would remove the diversity that distinguishes different examples of

the same class. We balance diversity and domain-invariance by interpolating $z(s)$ with the corresponding *class prototype representation*. We define the class prototype representation $r_c$ as the average *semantic* representation over examples belonging to class $c$ regardless of domain:

$$r_c = \frac{1}{n_c} \sum_{i=1}^{n_c} z(s_i) = \frac{1}{n_c} \sum_{i=1}^{n_c} \frac{s_i - \mu(s_i)}{\sigma(s_i)}. \tag{5}$$

For each example $(x, y, d)$ with $y = c$, we obtain the prototype-enhanced semantic representation by linearly interpolating $z(s)$ with the corresponding class prototype $r_c$:

$$z'(s) = \lambda_c z(s) + (1 - \lambda_c) r_c, \tag{6}$$

where $\lambda_c \sim \text{Beta}(\alpha_c, \alpha_c)$ is the interpolation coefficient. By applying this class prototype-based interpolation strategy, we are capable of capturing invariant knowledge and keeping the diversity of instance-level semantic representation when swapping information.

We also desire that the disentangled $\mu(s)$ and $\sigma(s)$ (Eqn. (2)) contain only domain-related nuisance information. However, for similar reasons as with $z(s)$, they may still contain some class-related semantic information which we would like to remove by "averaging out." In this case, we remove semantic information by averaging over examples from *different classes* within the *same domain*:

$$u_d = \frac{1}{n_d} \sum_{i=1}^{n_d} \mu(s_i), \; v_d = \frac{1}{n_d} \sum_{i=1}^{n_d} \sigma(s_i), \tag{7}$$

where $n_d$ represents the number of training examples in domain $d$. Then, for each example, we linearly interpolate its domain-associated nuisances with the above class-agnostic nuisances as:

$$\mu'(s) = \lambda_d \mu(s) + (1 - \lambda_d) u_d, \; \sigma'(s) = \lambda_d \sigma(s) + (1 - \lambda_d) v_d, \tag{8}$$

where the interpolation ratio is $\lambda_d \sim \text{Beta}(\alpha_d, \alpha_d)$. In practice, we update the class prototype $r_c$ and domain-agnostic nuisances $u_d$ and $v_d$ with momentum updating, where we denote the values of $r_c$, $u_d$, $v_d$ at epoch $t$ as $r_c^{(t)}$, $u_d^{(t)}$ and $v_d^{(t)}$, respectively.

By replacing the original semantic representation and domain-associated nuisances in Eqn. (4) with the prototype-guided ones, we obtain the enhanced augmented representation as follows:

$$\tilde{s}' = \sigma'(s_j) z'(s_i) + \mu'(s_j), \; \tilde{y}' = y_i. \tag{9}$$

Finally, we replace the original training data with the augmented ones and reformulate the optimization process in Eqn. (1) as:

$$\min_{\theta} \mathbb{E}_{(x_i, y_i), (x_j, y_j) \sim P^{tr}} [\ell(f_\theta^{L-r}(\tilde{s}'), \tilde{y}')], \tag{10}$$

where $f^{L-r}$ represents the post-layers after layer $r$. It is also worthwhile to point out that TALLY can be incorporated into any kinds of class-imbalanced losses (e.g., Focal, LDAM). We summarize the overall framework of TALLY in Algorithm 1.

## 4 Experiments

In this section, we conduct extensive experiments to answer the following questions: **Q1**: How does TALLY perform relative to prior invariant learning and single-domain long-tailed learning approaches under subpopulation shift and domain shift? **Q2**: Since it is straightforward to combine invariant learning with imbalanced data strategies, how does TALLY compare with such combinations? **Q3**: What affect does incorporating the prototype representation (Eqn. (9)) have, in comparison with naive representation swapping (Eqn. (4))? **Q4**: Can TALLY produce models with greater domain invariance?

We compare TALLY to two categories of algorithms. The first category includes single-domain long-tailed learning methods such as Focal (Lin et al., 2017), LDAM (Cao et al., 2019), CRT (Kang et al., 2020), MiSLAS (Zhong et al., 2021), RIDE (two experts) (Wang et al., 2020a), PaCo (Cui et al., 2021), and

Table 1: Results of subpopulation shifts and domain shifts on synthetic data. Domain-class balanced accuracy is reported. See full table with standard deviation in Appendix D.3. We bold the best results and underline the second best results. OH-LT and DN-LT represent OfficeHome-LT and DomainNet-LT, respectively.

| | Subpopulation Shift | | | | Domain Shift | | | |
|---|---|---|---|---|---|---|---|---|
| | VLCS-LT | PACS-LT | OH-LT | DN-LT | VLCS-LT | PACS-LT | OH-LT | DN-LT |
| ERM | 73.33% | 90.40% | 61.07% | 44.33% | 67.62% | 76.27% | 51.95% | 33.21% |
| Focal | 74.83% | 90.44% | 62.57% | 47.35% | 69.38% | 75.29% | 54.03% | 35.23% |
| LDAM | 73.83% | 90.91% | 63.57% | 46.71% | 69.41% | 77.53% | 53.60% | 34.42% |
| CRT | 73.83% | 89.17% | 61.92% | 47.37% | 65.67% | 73.82% | 53.62% | 36.14% |
| MiSLAS | 71.83% | 90.99% | 61.38% | 49.15% | 68.64% | 77.94% | 52.86% | 36.18% |
| Remix | 74.16% | 90.83% | 61.59% | 47.56% | 67.71% | 75.25% | 51.43% | 35.14% |
| RIDE | 74.33% | 90.48% | 63.27% | 47.51% | 69.29% | 77.41% | 53.75% | 35.26% |
| PaCo | 73.67% | 91.31% | 63.03% | 48.26% | 69.05% | 76.79% | 53.98% | 35.76% |
| IRM | 50.50% | 65.24% | 45.48% | 35.57% | 48.32% | 52.60% | 42.34% | 28.19% |
| GroupDRO | 72.50% | 89.80% | 59.79% | 43.86% | 69.18% | 76.75% | 51.12% | 32.54% |
| CORAL | 71.67% | 88.22% | 59.10% | 43.92% | 66.54% | 75.62% | 50.74% | 33.44% |
| LISA | 74.67% | 90.08% | 57.39% | 43.17% | 66.42% | 74.47% | 48.22% | 34.99% |
| MixStyle | 74.30% | 91.55% | 62.26% | 43.59% | 67.75% | 79.78% | 52.47% | 33.71% |
| DDG | 73.00% | 89.60% | 58.80% | 44.46% | 68.38% | 75.97% | 51.07% | 33.94% |
| BODA | 74.83% | 91.03% | 62.79% | 47.61% | 69.63% | 78.81% | 53.32% | 35.85% |
| **TALLY** (ours) | **76.83%** | **92.38%** | **67.00%** | **50.15%** | **70.60%** | **81.55%** | **55.69%** | **36.45%** |

Remix (Chou et al., 2020). The second category includes approaches for improving robustness to distribution shift: IRM (Arjovsky et al., 2019), GroupDRO (Sagawa et al., 2020), LISA (Yao et al., 2022), MixStyle (Zhou et al., 2020b), DDG (Zhang et al., 2022), and BODA (Yang et al., 2022), where BODA is a work studying multi-domain long-tailed learning by adding regularizer on domain-class pairs. Follow Yang et al. (2022), we use a ResNet-50 for all algorithms, and detail the baselines and evaluation metrics in Appendix C. All hyperparameters are selected via cross-validation.

## 4.1 Evaluation on Long-Tailed Variants of Domain Generalization Benchmarks

**Datasets.** Many standard domain-generalization benchmarks are not long-tailed, while standard imbalanced-classification datasets tend to be in a single-domain. We curate four *multi-domain long-tailed* datasets by modifying four existing domain-generalization benchmarks: VLCS (Fang et al., 2013), PACS (Li et al., 2017), OfficeHome (Venkateswara et al., 2017), and DomainNet (Peng et al., 2019). We modify the prior datasets by removing training examples so that each domain has a long-tailed label distribution (overall imbalance ratio: 50) and call the resulting datasets **VLCS-LT**, **PACS-LT**, **OfficeHome-LT**, and **DomainNet-LT**. See Appendix D.1 for more details.

**Evaluation Setup.** We evaluate performance under both subpopulation and domain shifts. In subpopulation shift, the test set is balanced across domains and classes, which means that each domain-class pair contains the same number of test examples. In domain shift, we use the classical domain generalization setting (Zhang et al., 2022). More specifically, we alternately use one domain as the test domain, and the rest as the training domains. Results are averaged over all combinations. Appendix D.1. detail the statistics and training class distribution for each multi-domain long-tailed dataset. The hyperparameters $\alpha_c$ and $\alpha_d$ in the Beta distribution are set to 0.5 and the warm start epoch $T_0$ is set to 7. We list all hyperparameters in Appendix D.2.

**Results on Subpopulation Shifts.** The overall performance of TALLY and prior methods for tackling subpopulation shift is reported in Table 1 (left). For subpopulation shift, we report the average performance over all domains and the full results are presented in Table 7 of Appendix D.3. The following key observations can be made according to Table 1 (left). We first observe that most single-domain re-weighting approaches (e.g., Focal, LDAM) consistently outperform multi-domain learning approaches (e.g., GroupDRO, CORAL) in most scenarios, indicating that imbalances in classes are probably more detrimental than imbalances in

domains. This observation is not surprising, since all domains are observed in during training and testing in subpopulation shift problems. Even so, TALLY consistently outperforms all methods with 7.29% error decreasing, verifying its effectiveness in improving the robustness to subpopulation shifts. It is particularly noteworthy that TALLY shows superior performance compared with BODA – an invariant learning approach to deal with multi-domain long-tailed learning. The results indicate that designing regularizers that are suitable for datasets from diverse domains can be challenging. Instead, balanced augmentation are capable of improving the robustness by transferring domain nuisances between examples.

In addition, Figure 4 shows performance broken down by class size for OfficeHome-LT and DomainNet-LT, where we split all classes into five levels according to their cardinality. We compare TALLY with ERM, and four strongest baselines (LDAM, CRT, MiSLAS, BODA). The results show that TALLY's performance improvements arise from larger improvements on smaller classes rather than performance improvements across the board, hence indicating that it is particularly well-suited for class-imbalanced problems.

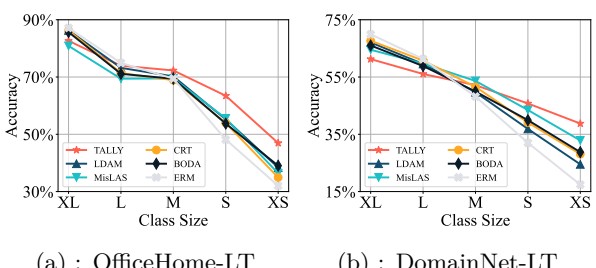

(a) : OfficeHome-LT          (b) : DomainNet-LT

Figure 4: Performance w.r.t. Class Size. We split all classes into five levels. XL and XS represent the largest and smallest classes, respectively.

**Results of Domain Shifts.** Table 1 (right) shows the domain shift results. We first find that ERM works relatively well compared to invariant learning approaches in most cases. This is expected since we evaluate the performance on unseen domains and similar observations have been reported from prior domain shift benchmarks (Koh et al., 2021). Second, single-domain long-tailed learning methods boost the performance of ERM in most cases, showing that class-imbalance is still an important issue in domain shift. Even so, as with the subpopulation shift setting, TALLY consistently outperforms prior approaches, indicating its efficacy in enhancing robustness to domain shift. Finally, we also provide the comparison on the standard version in Appendix G and TALLY also achieves comparable results compared with state-of-the-art methods.

## 4.2 Evaluation on Naturally Imbalanced Data

**Datasets and Evaluation Protocol.** To further evaluate TALLY and prior methods, we study two multi-domain datasets that are naturally imbalanced: Terra Incognita (TerraInc) (Beery et al., 2018) and iWildCam (Beery et al., 2020), both of which aim to classify wildlife across different camera traps. The examples are collected from different camera traps (domains) and the observed frequency of each species is naturally uneven, leading to an imbalanced class distribution. More details of these datasets and class distribution are described in Appendix E.1. Here, we consider generalization to new camera traps, i.e. to new domains. In TerraInc, the number of training, validation and test domains are 10, 5, 5, respectively. For iWildCam, we follow the same training, validation, and test splits as used in the WILDS benchmark (Koh et al., 2021). To better capture performance on rare species, we use macro F1 score as the primary evaluation metric following Koh et al. (2021), but we also report average accuracy. We list all hyperparameters in Appendix E.2.

Table 2: Results of domain shifts on real-world data. We report the full results in Appendix E.3.

|  | TerraInc | | iWildCam | |
| --- | --- | --- | --- | --- |
|  | Macro F1 | Acc | Macro F1 | Acc |
| ERM | 42.35% | 54.81% | 32.0% | 69.0% |
| Focal | 43.54% | 56.62% | _33.2%_ | 74.7% |
| LDAM | 44.29% | 57.22% | 32.7% | **75.2%** |
| CRT | 43.09% | 58.27% | 32.5% | 67.3% |
| MiSLAS | 40.68% | 52.96% | 30.5% | 59.8% |
| Remix | 43.72% | _58.40%_ | 28.4% | 65.8% |
| RIDE | 44.03% | 57.89% | 32.8% | 70.1% |
| PaCo | 43.40% | 57.35% | 31.9% | 72.6% |
| IRM | 31.17% | 49.27% | 15.1% | 59.8% |
| GroupDRO | 42.22% | 56.43% | 23.9% | 72.7% |
| CORAL | _45.43%_ | 58.10% | 32.8% | 73.3% |
| LISA | 39.27% | 54.92% | 27.6% | 64.9% |
| MixStyle | 44.73% | 57.55% | 32.4% | _74.9%_ |
| DDG | 40.47% | 53.61% | 29.8% | 69.7% |
| BODA | 44.47% | 57.52% | 32.9% | 70.5% |
| **TALLY (ours)** | **46.23%** | **59.89%** | **34.4%** | 73.4% |

**Results.** We report the results over all test domains in Table 2. The conclusions are largely consistent with the results from Sec. 4.1, where TALLY consistently improves the performance over baselines and enhances

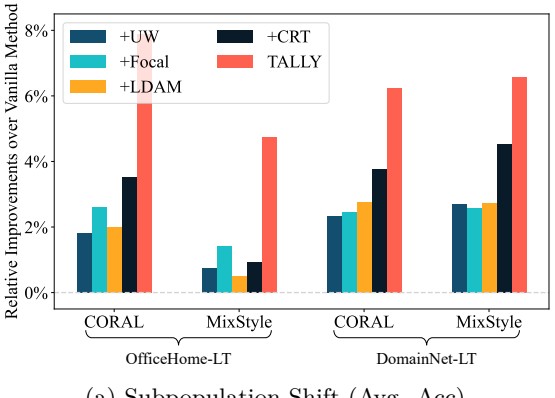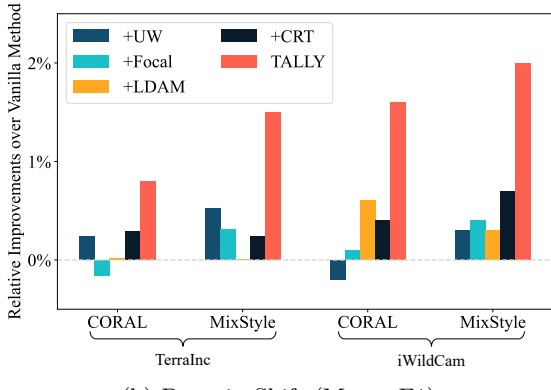

(a) Subpopulation Shift (Avg. Acc)        (b) Domain Shift (Macro F1)

Figure 5: Comparison between TALLY and variants of two domain generalization approaches (CORAL, MixStyle), where we replace the losses of them with class re-weighting or re-sampling ones.

the robustness of multi-domain long-tailed learning. Additionally, data interpolation based invariant learning approaches (e.g., LISA) hurt performance compared with ERM. This is not a surprise because examples from large classes dominate the interpolation, which essentially exacerbate the class imbalance (See Appendix B for more discussion). The superiority of TALLY over prior augmentation techniques is further evidence of the effectiveness of balanced augmentation.

### 4.3 Can we simply combine invariant learning approaches with long-tailed learning techniques?

To further understand the performance gains of TALLY, we investigate whether combining existing invariant learning and long-tailed learning approaches can tackle multi-domain long-tailed distribution shifts. Specifically, we incorporate four up-weighting or up-sampling approaches (UW, Focal, LDAM, CRT) with two representative invariant learning methods (CORAL, MixStyle). We report the relative improvement of each combination over the vanilla methods in Figure 6. Here, we use Officehome-LT and DomainNet-LT to evaluate subpopulation shift and TerraInc and iWildCam to evaluate domain shift performance. We see that applying loss up-weighting or up-sampling approaches on performant invariant learning approaches does improve their performance, as evidenced by Figure 5. Nonetheless, the consistent improvements from TALLY indicate the importance of considering domain-class pair information to achieve balanced augmentation.

### 4.4 Analysis

#### 4.4.1 How do prototypes benefit invariant learning?

We analyze the effects of prototypes in alleviating domain-associated nuisances. Specifically, we compare TALLY with three variants: (1) without using any prototype information (None); (2) only applying class prototype (C Only); (3) only applying class-agnostic nuisances (D Only). We report the results in Figure 6 (full results in Appendix F.3). We observe that adding class prototype does improve the performance. The class-agnostics domain factors also benefits the performance to some extent. In summary, TALLY outperforms its variants, verifying the effectiveness of prototype representation in mitigating nuisances.

#### 4.4.2 Does TALLY lead to stronger domain invariance?

We analyze and compare the domain invariance of classifiers trained by ERM, TALLY, and other invariant learning approaches. Following (Yang et al., 2022; Yao et al., 2022), we measure the lack of domain invariance as the accuracy of domain prediction ($I_{acc}$) and as the pairwise divergence of unscaled logits ($I_{kl}$). Specifically, for the accuracy of domain prediction, we perform logistic regression on top of the unscaled logits to predict the domain. For the pairwise divergence, we use kernel density estimation to estimate the probability density function $P(h^{c,d})$ of logits from domain-class pair $(c, d)$ and calculate the KL divergence of the distribution of logits from different pairs. Formally, $I_{kl}$ is defined as $I_{kl} = \frac{1}{|\mathcal{C}||\mathcal{D}|^2} \sum_{c \in \mathcal{C}} \sum_{d', d \in \mathcal{D}} \text{KL}(P(h^{c,d})|P(h^{c,d'}))$. We

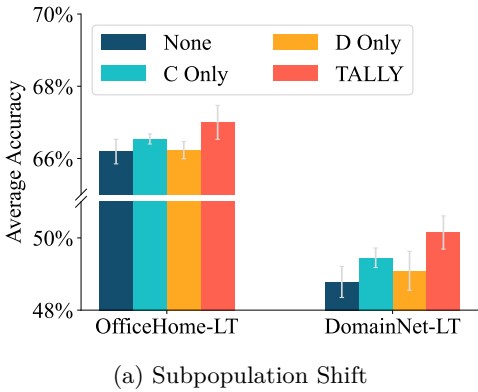

(a) Subpopulation Shift

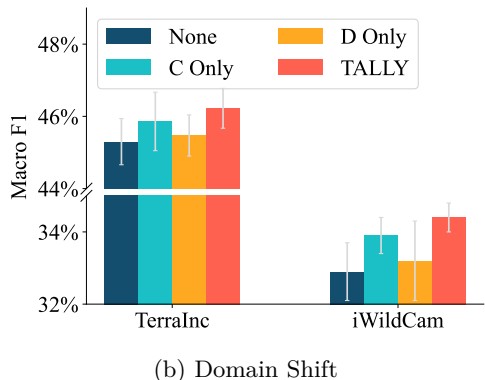

(b) Domain Shift

Figure 6: Analysis of prototype-guided invariant learning. C Only and D Only represent only using class prototype representation or class-agnostic domain factors, respectively.

Table 3: Invariance Analysis of TALLY.

| Model | OfficeHome-LT | | DomainNet-LT | |
|-------|---------------|---|--------------|---|
| | $I_{acc} \downarrow$ | $I_{kl} \downarrow$ | $I_{acc} \downarrow$ | $I_{kl} \downarrow$ |
| ERM | 46.35% | 2.030 | 70.00% | 4.852 |
| MixStyle | 44.42% | 2.169 | 67.11% | 5.661 |
| CORAL | 42.21% | 1.248 | 66.79% | 4.593 |
| BODA | 40.10% | 2.052 | 65.15% | 6.810 |
| **TALLY** | **39.52%** | **1.179** | **63.80%** | **3.956** |

Table 4: Comparison between sampling strategies.

| | OH-LT | DN-LT |
|-------|-------|-------|
| Balanced Sampling | 65.03% | 49.35% |
| **TALLY**(Selective) | **67.00%** | **50.15%** |

| | TerraInc | iWildCam |
|-------|----------|----------|
| Balanced Sampling | 44.79% | 33.1% |
| **TALLY**(Selective) | **46.23%** | **34.4%** |

report the results of Officehome-LT and DomainNet-LT in Table 3. Smaller $I_{acc}$ and $I_{kl}$ values indicate more invariant representations with respect to the labels. The results show that TALLY does lead to greater domain-invariance compared to prior invariant learning approaches (e.g., BODA).

### 4.4.3 Analysis of Sampling Strategies.

Finally, we compare the proposed selective balanced sampling in TALLY with domain-class balanced sampling. As discussed in Sec. 3.2, for an example pair $(x_i, y_i, d_i)$ and $(x_i, y_j, d_j)$, selective balanced sampling gets $y_i \sim \text{Uniform}(\mathcal{C})$ and $d_j \sim \text{Uniform}(\mathcal{D})$, while traditional balanced sampling get $(y_i, d_i), (y_j, d_j) \sim \text{Uniform}(\mathcal{C}, \mathcal{D})$. The results of subpopulation shifts in OfficeHome-LT, DomainNet-LT and of domain shifts (Macro-F1) in TerraInc, iWildCam are reported in Table 4 (see Appendix F.4 for full results), indicating the effectiveness of selective balanced sampling in transferring knowledge over domains and classes.

### 4.4.4 Analysis of Prototype Construction Methods

To further investigate the sensitivity of the method to the quality of class prototype, we have investigated more complicated prototype construction strategies. Specifically, we leveraged DeepSets transformation (Ye et al., 2020) to model the correlations between different prototypes. We report the results in Table 5. Here, using DeepSets to advance prototypes is denoted as Advanced Prototype.According to the results in Table R1, we observe that the vanilla prototype construction strategy shows better performance compared to other strategies. The potential reason is that it might be hard to train a well-performed transformation that can accurately capture the correlations behind prototypes. We thus use the vanilla prototype construction method.

## 5 Related Work

**Long-Tailed Learning.** Training a well-performed machine learning model on class-imbalanced data has been widely studied and a lot of approaches have been proposed, including over-sampling minority classes or

Table 5: Comparison with different prototype construction strategies.

|  | OH-LT | DN-LT | TerraInc | iWildCam |
|---|---|---|---|---|
| Advanced Prototype | $65.12 \pm 0.21\%$ | $49.05 \pm 0.70\%$ | $44.12 \pm 0.44\%$ | $32.7 \pm 0.3\%$ |
| **TALLY** | $\mathbf{67.00 \pm 0.47\%}$ | $\mathbf{50.15 \pm 0.46\%}$ | $\mathbf{46.23 \pm 0.56\%}$ | $\mathbf{34.4 \pm 0.4\%}$ |

under-sampling majority classes (Chawla et al., 2002; Estabrooks et al., 2004; Kang et al., 2020; Liu et al., 2008; Zhang & Pfister, 2021), adjusting loss functions or logits for different classes during training (Cao et al., 2019; Cui et al., 2019; Hong et al., 2021; Jamal et al., 2020; Lin et al., 2017), transferring knowledge from head classes to tail classes (Wang et al., 2017; Liu et al., 2020; Yin et al., 2019; Zhou et al., 2022), directly augmenting tail classes (Chou et al., 2020; Ye et al., 2021; Zhong et al., 2021), and ensembling models with different sampling or loss weighting strategies (Xiang et al., 2020; Zhou et al., 2020a). Unlike single-domain imbalanced learning, Yang et al. (2022) targets on the multi-domain imbalanced learning scenario by encouraging invariant representation learning with a domain-class calibrated regularizer. However, BODA focuses on subpopulation shift with domain-specific imbalanced distribution, while the overall distribution among all classes are relatively balanced. TALLY instead studies more kinds of distribution shifts with conceptually different direction to alleviate domain-associated nuisances via balanced augmentation.

**Domain Generalization and Out-of-Distribution Robustness.** To improve out-of-distribution robustness, one line of works aims to learn domain-invariant representations by 1) minimizing the discrepancy of feature representations across all training domains (Li et al., 2018; Sun & Saenko, 2016; Zhou et al., 2020b); 2) leveraging domain augmentation methods to generate more training domains and improve the consistency among domains (Shu et al., 2021; Wang et al., 2020b; Xu et al., 2020; Yan et al., 2020; Yue et al., 2019; Zhou et al., 2020c); 3) disentangling feature representations to semantic and domain-varying ones and minimizing the semantic differences across training domains (Robey et al., 2021; Zhang et al., 2022). Another line of works focuses on learning invariant predictors with regularizers, including minimizing the variances of risks across domains (Krueger et al., 2021), encouraging a predictor that performs well over all domains (Ahuja et al., 2021; Arjovsky et al., 2019; Guo et al., 2021; Khezeli et al., 2021). Apart from explicitly involving regularizers, data augmentation is another promising approach for learning invariant predictors (Yao et al., 2022; Zhou et al., 2020b), which provides a greater degree of flexibility in regularizing the model. Unlike previous augmentation methods that require sufficient training examples for each class to learn invariance (see detailed discussion in Appendix B), TALLY tackles the class-imbalanced issue in domain generalization and employs a domain-balanced augmentation strategy to learn class-unbiased invariant representation.

# 6 Conclusion and Discussion

In this paper, we investigate multi-domain imbalanced learning, a natural extension of classical single-domain imbalanced learning. We propose a novel balanced augmentation algorithm called TALLY to achieve robust imbalanced learning that can overcome distribution shifts. To generate more examples, TALLY introduces a prototype enhanced disentanglement procedure for separating semantic and nuisance information. TALLY then mixes the enhanced semantic and domain-associated nuisance information among examples. The results on four synthetic and two real-world datasets demonstrate its effectiveness over existing imbalanced classification and invariant learning techniques.

While TALLY demonstrates promising results, an intriguing future direction includes both integrating TALLY with additional disentanglement techniques and applying it to a wider range of data types, such as text data and drug data. Additionally, TALLY experiences a slight decline in performance when dealing with larger classes under subpopulation shifts. In the future, we can enhance TALLY's performance through ensemble models tailored for different class sizes.

# Acknowledgement

We thank Pang Wei Koh, Yoonho Lee, Sara Beery, and members of the IRIS lab for the many insightful discussions and helpful feedback. This research was funded in part by Apple, Intel, Juniper Networks, and

JPMorgan Chase & Co. Any views or opinions expressed herein are solely those of the authors listed, and may differ from the views and opinions expressed by JPMorgan Chase & Co. or its affiliates. This material is not a product of the Research Department of J.P. Morgan Securities LLC. This material should not be construed as an individual recommendation for any particular client and is not intended as a recommendation of particular securities, financial instruments or strategies for a particular client. This material does not constitute a solicitation or offer in any jurisdiction. CF is a CIFAR fellow.

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

# A  Additional Information for TALLY

## A.1  Additional Discussion of Selective Balanced Sampling

In this section, we detail our explanation about Figure 3 and show why selective balanced sampling is a better strategy in TALLY. In Figure 3, there are three domain-class groups: (Fan, Clipart), (Fan, Art), (Computer, Clipart) and the number of examples for each group is 1, 13, 83, respectively. We specifically focus on augmenting examples from the minority group (i.e., (Computer, Clipart) group). In this case, the samples $(x_i, y_i, d_i)$ and $(x_j, y_j, d_j)$ need to contain class semantic information (i.e., Fan) and domain information (i.e., Clipart) in the representation augmentation module, respectively.

The balanced sampling under the multi-domain long-tailed classification problem is to up-sample examples from minority domain-class groups. Concretely, the label $y_i$ and domain $d_i$ for each example $(x_i, y_i, d_i)$ are jointly sampled from Uniform$(\mathcal{C}, \mathcal{D})$. If we employ original balanced sampling in the representation augmentation module of TALLY, then $(y_i, d_i)$ Uniform$(\mathcal{C}, \mathcal{D})$, $(y_j, d_j)$ Uniform$(\mathcal{C}, \mathcal{D})$. To augment the (Fan, Clipart) group, the class semantic information from $(x_i, y_i, d_i)$ has a $1/2$ probability to be obtained from the original (**Fan**, Clipart) group, and a $1/2$ probability to be obtained from the (**Fan**, Art) group. Similarly, the domain information has a $1/2$ probability to be obtained from the original (Fan, **Clipart**) group, and a $1/2$ probability to be obtained from the (Computer, **Clipart**) group. Thus, examples from the minority group (i.e., Fan, Clipart) will be repeatedly sampled during the augmentation process and this is what we do not expect.

Instead, for selective balanced sampling, the label $y_i$ of example $i$ is uniformly sampled from all classes (i.e., $y_i$ Uniform$(\mathcal{C})$), and the domain $d_j$ or example $j$ is uniformly sampled from all domains (i.e., $d_i$ Uniform$(\mathcal{D})$). Thus, to augment the (Fan, Clipart) group, the class semantic information from example $(x_i, y_i, d_i)$ has a $1/14$ probability to be obtained from the original (**Fan**, Clipart) group and a $13/14$ probability to be obtained from the (**Fan**, Art) group because we do not consider domain information in sampling example $(x_i, y_i, d_i)$. Similarly, for selective balanced sampling, the domain information has a $1/84$ probability to be obtained from the original (Fan, **Clipart**) group, and a $83/84$ probability to be obtained from the (Computer, **Clipart**) group.

To sum up, using selective balanced sampling can provide more diverse and effective knowledge transfer.

## A.2  Algorithm of the Testing Stage of TALLY

In this section, we summarize the testing stage of TALLY in Alg. 2. It is worthwhile to notice that the representation augmentation module is only used during the training stage.

---

**Algorithm 2** TALLY Testing Process

---

**Require:** model $f_{\theta^*}(\cdot)$ with learned parameter $\theta^*$; Test Dataset $\mathcal{D}^{ts}$.
1: Feed all testing examples $\{(x_i, y_i, d_i)\}_{i=1}^{n^{ts}}$ to model $f_{\theta^*}(\cdot)$ and get the predicted label of each example $(x, y, d)$ as $\hat{y} = f_{\theta^*}(x)$
2: Evaluate and report the performance based on the predicted values and the groundtruth.

---

# B  Additional Discussion of Related Works

In this section, we provide an additional discussion of related works. Specifically, we would like to point out why data interpolation-based domain generalization approaches (e.g., LISA (Yao et al., 2022), mixup) can not benefit the performance when encountering the long-tailed distribution. Take LISA as an example, we adopt intra-label LISA in this paper, which is more suitable for domain shift as mentioned in (Yao et al., 2022). Intra-label LISA learns domain-invariant predictors by interpolating examples with the same label but from different domains, which can probably aggravate the label imbalance issue. We provide a simple example to explain why LISA changes the label distribution. Assume we have two classes and two domains, the ratio of training examples between four domain-class pairs in training set is: $(y_1, d_1) : (y_1, d_2) : (y_2, d_1) : (y_2, d_1) = 100 : 200 : 80 : 5$. The label imbalance ratio is $300 : 85 = 3.52$. To apply LISA, we can essentially have $100 \times 200 = 20000$ example pairs in class 1 $(y_1)$, and $80 \times 5 = 400$ example pairs in class 2 $(y_2)$, which roughly leads to a new label imbalance ratio $20000 : 400 = 50 > 3.52$. The comparison between the number of example pairs for interpolation can not precisely reflect the training label imbalance ratio in LISA, but it shows that LISA changes the label distribution to some extent.

# C  Additional Details of General Experimental Setups

## C.1  Detailed Baseline Descriptions

In this paper, we compare TALLY with two types of approaches: long-tailed classification methods and invariant learning approaches. We detail these methods here:

**Long-tailed Classification Methods.** We compare TALLY with Focal Lin et al. (2017), LDAM Cao et al. (2019), CRT Kang et al. (2020), MiSLAS Zhong et al. (2021), and Remix Chou et al. (2020). Here, Focal and LDAM up-weight the loss for minority classes. CRT uses up-sampling strategy to fine-tune the classifier. MiSLAS and Remix modify the vanilla mixup Zhang et al. (2018) and make it suitable to long-tailed distribution.

**Invariant Learning.** We further compare TALLY with invariant learning approaches, i.e., IRM Arjovsky et al. (2019), GroupDRO Sagawa et al. (2020), LISA Yao et al. (2022), MixStyle Zhou et al. (2020b), DDG Zhang et al. (2022), and BODA Yang et al. (2022). IRM learns invariant predictors that perform well across different domains. GroupDRO optimizes the worst-domain loss. LISA cancels out domain-associated information by mixing examples with the same label but different domains. MixStyle decomposes the feature representation into content information and style information. It then mixes the style information and generates new examples. Unlike MixStyle, TALLY generates examples of minority classes or domains, and uses prototypes to improve the model robustness, which is more suitable for long-tailed multi-domain learning. DDG uses an extra network to disentangle original examples and generate more. Finally, BODA is a concurrent work for long-tailed multi-domain learning with an explicit regularizer. Unlike BODA, TALLY studies a conceptually different direction to cancel out domain-associated nuisances by domain-class balanced augmentation, leading to stronger empirical performance.

## C.2  Detailed Evaluation Metrics

In this section, we detail our evaluation metrics. In synthetic data, to emphasize the performance on minority classes and domains, we use domain-class balanced accuracy as the evaluation metric, where the number of test examples is the same across every class within every test domain. In real-world data, follow Koh et al.

(2021), we use Macro F1 over all classes and average accuracy to evaluate the performance, where the Macro F1 is served as the primary metric to emphasize the performance on rare classes.

# D Additional Results of Synthetic Data

## D.1 Detailed Dataset Description

**VLCS-LT** contains examples from 4 different domains, including Caltech101, LabelMe, SUN09, VOC2007. To create the long-tailed class distribution, we modify the original dataset by removing training examples. The dataset contains 5 classes with 6,361 images of dimension (224, 224, 3). The long-tailed training distribution is visualized in Figure 7a. In subpopulation shift, the number of examples of each class per domain for validation and testing is 5, 10, respectively.

**PACS-LT** includes 3,097 images collected from 4 domains (Art painting, Cartoon, Photo, Sketch) and 7 classes. Similar to VLCS-LT, we construct PACS-LT with long-tailed training distribution illustrated in Figure 7b. The validation set size and test set size of each class per domain in subpopulation shift are 15 and 30 respectively.

**OfficeHome-LT** is built upon the original OfficeHome dataset, including 3280 images of 65 classes collected from four domains – Art, Clipart, Product, Real. The long-tailed training distribution is shown in Figure 7c and the number of examples for each class per domain in validation and test sets are 4, 8, respectively.

**DomainNet-LT.** Similar to the other three datasets, DomainNet-LT covers 173,200 examples from Sketch, Infograph, Painting, Quickdraw, Real, Clipart. There are 345 classes in DomainNet-LT. In subpopulation shift, the number of examples of each class per domain is 3, 6, respectively. We illustrate the long-tailed training distribution in Figure 7d.

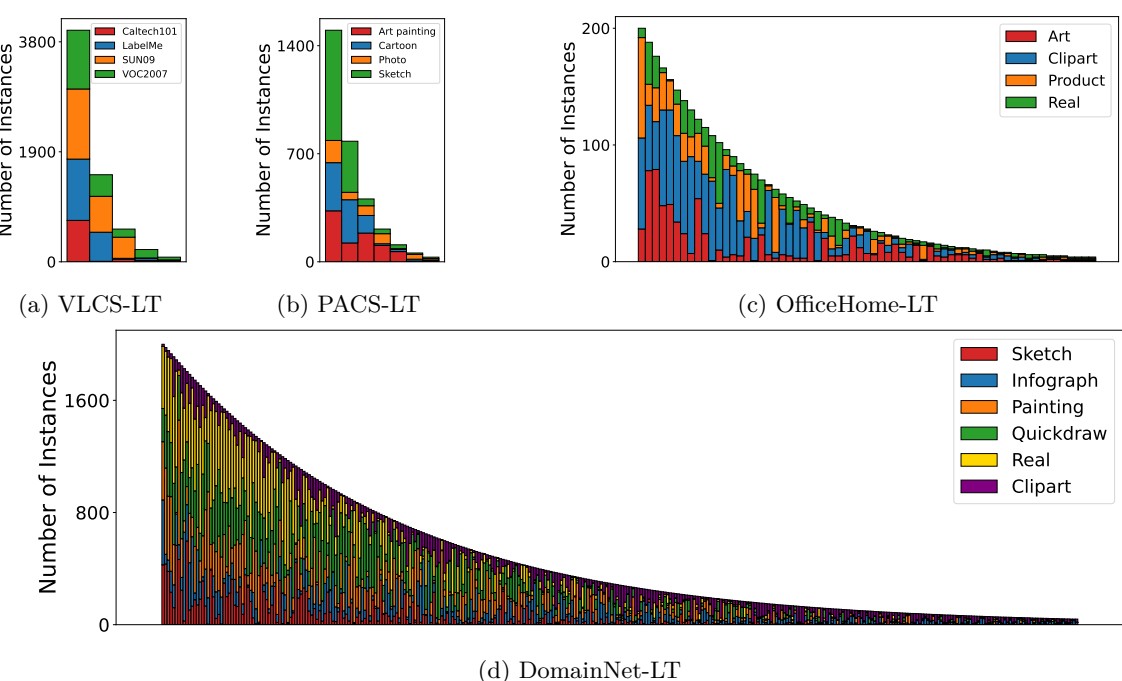

Figure 7: Long-tailed training distributions for all synthetic datasets. Here, the x-axis represents sorted class indices.

### D.2 Detailed Experimental Setups and Hyperparameters

In this section, we detail how we split the training and test set in synthetic datasets under both subpopulation shifts and domain shifts. In subpopulation shift, the training distribution for each domain is a long-tailed distribution and the test is domain-class balanced distribution, i.e., the number of examples in each domain-class pair is the same. In terms of domain shifts in synthetic datasets, following (Peng et al., 2019; Shi et al., 2021), we hold out one domain as the testing domain and use the rest domains as training. All baselines and TALLY use the same evaluation protocol.

We list the hyperparameters in Table 6 for the above four synthetic datasets.

Table 6: Hyperparameters for experiments on synthetic data.

| Hyperparameters | VLCS-LT | PACS-LT | OfficeHome-LT | DomainNet-LT |
|---|---|---|---|---|
| Learning Rate | 1e-5 | 1e-5 | 3e-5 | 3e-5 |
| Weight Decay | 1e-6 | 1e-6 | 1e-6 | 1e-6 |
| Batch Size | 18 | 18 | 18 | 18 |
| Epochs | 15 | 15 | 15 | 15 |
| Steps | 200 | 500 | 500 | 1000 |
| Warm Start Epochs | 7 | 7 | 7 | 7 |
| $\gamma$ in feat. estimation | 0.8 | 0.8 | 0.8 | 0.8 |
| class prototype mixup parameter $\alpha_c$ | 0.2 | 0.5 | 0.5 | 0.5 |
| domain prototype mixup parameter $\alpha_d$ | 0.2 | 0.5 | 0.5 | 0.5 |

### D.3 Full Results

The full results of subpopulation shift are reported in Table 7. In domain shift, we report the results of each domain for VLCS-LT, PACS-LT, OfficeHome-LT and DomainNet-LT in Table 8, 9, 10, 11, respectively. In the domain shift scenario of VLCS-LT, though TALLY only performs best in VOC2007 (VLCS), the results of TALLY is relatively more stable compared to other approaches, leading to the best averaged performance.

## E Additional Results of Real-world Data

### E.1 Detailed Dataset Description

**TerraInc.** Building upon the original Terra Incognita Beery et al. (2018), we select images from 10 classes and split the entire dataset to training, validation and test domains, which includes images from 38,042, 6,783, 7,303 camera traps, respectively.

**iWildCam** is a wildlife recognition datasets. It is a multi-class species classification, where the training data are collected from 243 domains and the test data includes images from 164 domains. We follow Koh et al. (2021) to split the data and construct training, validation and test sets.

### E.2 Detailed Experimental Setups and Hyperparameters

In this section, we detail how we split the training and test set in real-world datasets with domain shifts. Specifically, we follow Koh et al. (2021) and use the same split as they did for iWildCam, where a bunch of locations is selected for testing and the rest ones are used for training. For Terrainc, we adopt the same strategy to split training and test domains since Terrainc also focuses on wildlife recognition and the data distribution is similar to iWildCam. All baselines and TALLY use the same evaluation protocol.

We list the hyperparameters in Table 12 for both TerraInc and iWildCam datasets.

Table 7: Full results of subpopulation shifts on long-tailed variants of domain generalization benchmarks. The standard deviation is computed across three seeds.

| | | VLCS-LT | PACS-LT | OfficeHome-LT | DomainNet-LT |
|---|---|---|---|---|---|
| Avg. | ERM | $73.33 \pm 0.76\%$ | $90.40 \pm 0.88\%$ | $61.07 \pm 0.73\%$ | $44.33 \pm 0.14\%$ |
| | Focal | $\underline{74.83 \pm 0.29\%}$ | $90.44 \pm 0.06\%$ | $62.57 \pm 0.50\%$ | $47.35 \pm 0.09\%$ |
| | LDAM | $73.83 \pm 1.04\%$ | $90.91 \pm 0.15\%$ | $\underline{63.57 \pm 0.08\%}$ | $46.71 \pm 0.33\%$ |
| | CRT | $73.83 \pm 1.89\%$ | $89.17 \pm 1.47\%$ | $61.92 \pm 0.54\%$ | $47.37 \pm 0.83\%$ |
| | MiSLAS | $71.83 \pm 1.25\%$ | $90.99 \pm 0.90\%$ | $61.38 \pm 0.19\%$ | $\underline{49.15 \pm 0.69\%}$ |
| | Remix | $74.16 \pm 0.76\%$ | $90.83 \pm 0.77\%$ | $61.59 \pm 0.44\%$ | $47.56 \pm 0.25\%$ |
| | RIDE | $74.33 \pm 0.76\%$ | $90.48 \pm 0.32\%$ | $63.27 \pm 0.54\%$ | $47.51 \pm 0.83\%$ |
| | PaCo | $73.67 \pm 0.28\%$ | $91.31 \pm 0.58\%$ | $63.03 \pm 0.92\%$ | $48.26 \pm 0.22\%$ |
| | IRM | $50.50 \pm 8.18\%$ | $65.24 \pm 7.57\%$ | $45.48 \pm 4.30\%$ | $35.57 \pm 5.76\%$ |
| | GroupDRO | $72.50 \pm 0.50\%$ | $89.80 \pm 0.70\%$ | $59.79 \pm 0.43\%$ | $43.86 \pm 0.33\%$ |
| | CORAL | $71.67 \pm 0.28\%$ | $88.22 \pm 0.67\%$ | $59.10 \pm 0.20\%$ | $43.92 \pm 0.36\%$ |
| | LISA | $74.67 \pm 0.76\%$ | $90.08 \pm 0.45\%$ | $57.39 \pm 0.59\%$ | $43.17 \pm 0.53\%$ |
| | MixStyle | $74.30 \pm 1.04\%$ | $\underline{91.55 \pm 0.25\%}$ | $62.26 \pm 0.22\%$ | $43.59 \pm 0.57\%$ |
| | DDG | $73.00 \pm 1.63\%$ | $89.60 \pm 0.40\%$ | $58.80 \pm 0.57\%$ | $44.46 \pm 0.06\%$ |
| | BODA | $\underline{74.83 \pm 1.84\%}$ | $91.03 \pm 0.31\%$ | $62.79 \pm 0.45\%$ | $47.61 \pm 0.04\%$ |
| | **TALLY (ours)** | $\mathbf{76.83 \pm 1.04\%}$ | $\mathbf{92.38 \pm 0.26\%}$ | $\mathbf{67.00 \pm 0.47\%}$ | $\mathbf{50.15 \pm 0.46\%}$ |
| Worst | ERM | $52.67 \pm 2.31\%$ | $83.81 \pm 2.43\%$ | $54.48 \pm 0.89\%$ | $25.36 \pm 0.63\%$ |
| | Focal | $52.67 \pm 1.15\%$ | $84.44 \pm 0.81\%$ | $56.41 \pm 1.34\%$ | $27.68 \pm 0.13\%$ |
| | LDAM | $51.33 \pm 2.31\%$ | $85.24 \pm 1.03\%$ | $\underline{58.07 \pm 0.82\%}$ | $27.23 \pm 0.26\%$ |
| | CRT | $52.00 \pm 0.00\%$ | $83.02 \pm 1.12\%$ | $55.51 \pm 0.79\%$ | $27.55 \pm 0.49\%$ |
| | MiSLAS | $52.00 \pm 3.46\%$ | $86.03 \pm 0.90\%$ | $52.82 \pm 0.39\%$ | $\underline{29.42 \pm 0.15\%}$ |
| | Remix | $51.33 \pm 3.05\%$ | $86.98 \pm 0.59\%$ | $53.85 \pm 0.54\%$ | $28.13 \pm 0.99\%$ |
| | RIDE | $53.50 \pm 2.31\%$ | $85.71 \pm 0.82\%$ | $54.04 \pm 0.28\%$ | $28.16 \pm 0.60\%$ |
| | PaCo | $53.83 \pm 0.94\%$ | $\underline{87.14 \pm 0.46\%}$ | $56.15 \pm 0.71\%$ | $27.36 \pm 0.52\%$ |
| | IRM | $32.63 \pm 7.03\%$ | $59.38 \pm 5.93\%$ | $40.58 \pm 4.42\%$ | $20.48 \pm 3.71\%$ |
| | GroupDRO | $51.33 \pm 1.15\%$ | $83.02 \pm 0.59\%$ | $54.04 \pm 0.30\%$ | $25.02 \pm 0.73\%$ |
| | CORAL | $49.33 \pm 1.15\%$ | $81.59 \pm 0.81\%$ | $53.53 \pm 0.60\%$ | $24.50 \pm 0.68\%$ |
| | LISA | $53.33 \pm 1.15\%$ | $83.01 \pm 0.81\%$ | $49.04 \pm 0.40\%$ | $24.05 \pm 0.48\%$ |
| | MixStyle | $\underline{54.00 \pm 2.00\%}$ | $86.98 \pm 0.98\%$ | $55.19 \pm 1.10\%$ | $22.65 \pm 0.22\%$ |
| | DDG | $51.33 \pm 0.94\%$ | $82.70 \pm 2.38\%$ | $51.99 \pm 0.55\%$ | $24.35 \pm 0.20\%$ |
| | BODA | $\underline{54.00 \pm 2.83\%}$ | $85.08 \pm 1.37\%$ | $55.70 \pm 0.50\%$ | $26.94 \pm 0.44\%$ |
| | **TALLY (ours)** | $\mathbf{56.00 \pm 2.00\%}$ | $\mathbf{89.21 \pm 0.22\%}$ | $\mathbf{60.45 \pm 0.09\%}$ | $\mathbf{29.55 \pm 0.19\%}$ |

### E.3 Full Results

The full results on Real-world Data are reported in Table 13.

## F Additional Results of Analysis

In this section, we conduct two additional analysis to better understand how TALLY works. Then, we provide additional results of the analysis in the main paper.

### F.1 Fine-tuning ERM on Balanced Datasets

In this section, we conducted an additional experiment to compare TALLY with a simpler approach – training ERM on a imbalanced dataset and finetuning it on the corresponding balanced dataset. Here, we adopt three variants of balanced datasets for fine-tuning: (1) class-balanced dataset, where the number of examples is the

Table 8: Domain shift results on VLCS-LT.

|  | Caltech101 | LabelMe | SUN09 | VOC2007 | Avg |
|---|---|---|---|---|---|
| ERM | 92.39 ± 0.35% | 47.74 ± 1.13% | 59.79 ± 2.70% | 70.55 ± 1.51% | 67.62% |
| Focal | **97.12 ± 1.06%** | 48.83 ± 0.38% | 58.66 ± 2.31% | 72.91 ± 1.51% | 69.38% |
| LDAM | 95.55 ± 1.65% | 47.61 ± 1.12% | 61.34 ± 3.02% | 73.17 ± 1.33% | 69.41% |
| CRT | 92.39 ± 1.82% | 47.74 ± 1.52% | 55.10 ± 2.61% | 67.45 ± 1.54% | 65.67% |
| MiSLAS | 95.24 ± 1.51% | 47.00 ± 0.99% | 56.03 ± 1.29% | 76.28 ± 1.66% | 68.64% |
| Remix | 92.66 ± 1.42% | 48.77 ± 1.31% | 57.98 ± 2.62% | 71.45 ± 0.87% | 67.71% |
| RIDE | 95.70 ± 1.24% | 48.13 ± 0.58% | 59.62 ± 1.53% | 73.72 ± 1.19% | 69.29% |
| PaCo | 96.11 ± 0.98% | 49.48 ± 1.01% | 58.85 ± 2.04% | 71.76 ± 1.52% | 69.05% |
| IRM | 74.10 ± 2.67% | 37.07 ± 1.87% | 34.33 ± 1.77% | 47.78 ± 3.59% | 48.32% |
| GroupDRO | 93.79 ± 1.01% | 49.63 ± 1.09% | **62.25 ± 1.89%** | 71.06 ± 0.55% | 69.18% |
| CORAL | 93.94 ± 1.53% | 48.29 ± 1.08% | 56.12 ± 1.84% | 67.82 ± 1.43% | 66.54% |
| LISA | 90.28 ± 0.68% | 48.51 ± 1.58% | 58.82 ± 2.41% | 68.09 ± 1.53% | 66.42% |
| MixStyle | 96.58 ± 0.84% | 48.15 ± 1.20% | 58.82 ± 1.94% | 68.09 ± 1.98% | 67.75% |
| DDG | 95.46 ± 1.19% | 50.42 ± 1.45% | 57.44 ± 2.07% | 70.21 ± 1.33% | 68.38% |
| BODA | 95.60 ± 1.37% | **51.42 ± 1.31%** | 59.93 ± 1.97% | 71.57 ± 1.18% | 69.63% |
| **TALLY (ours)** | 95.22 ± 0.92% | 50.07 ± 1.17% | 60.13 ± 2.17% | **76.98 ± 0.57%** | **70.60%** |

Table 9: Domain shift results on PACS-LT.

|  | Art painting | Cartoon | Photo | Sketch | Avg |
|---|---|---|---|---|---|
| ERM | 80.41 ± 1.21% | 70.21 ± 1.14% | 94.46 ± 0.19% | 60.00 ± 5.04% | 76.27% |
| Focal | 80.92 ± 0.51% | 69.58 ± 0.64% | 93.81 ± 0.80% | 56.83 ± 2.04% | 75.29% |
| LDAM | 81.82 ± 1.14% | 71.64 ± 0.66% | 95.34 ± 0.32% | 61.30 ± 4.83% | 77.53% |
| CRT | 78.14 ± 0.99% | 67.17 ± 0.73% | 94.33 ± 0.78% | 55.62 ± 6.57% | 73.82% |
| MiSLAS | 81.31 ± 0.49% | 71.15 ± 0.28% | 93.51 ± 1.40% | 65.78 ± 2.13% | 77.94% |
| Remix | 82.79 ± 1.21% | 69.10 ± 1.13% | 92.09 ± 1.01% | 57.00 ± 4.13% | 75.25% |
| RIDE | 81.57 ± 0.57% | 71.64 ± 0.42% | 94.69 ± 0.49% | 61.74 ± 2.52% | 77.41% |
| PaCo | 80.86 ± 0.88% | 71.29 ± 0.28% | 95.45 ± 0.85% | 59.55 ± 3.02% | 76.79% |
| IRM | 51.87 ± 4.93% | 50.27 ± 5.77% | 69.11 ± 4.43% | 39.13 ± 9.65% | 52.60% |
| GroupDRO | 80.20 ± 0.57% | 70.61 ± 1.41% | 94.58 ± 0.90% | 61.61 ± 1.48% | 76.75% |
| CORAL | 77.60 ± 0.79% | 68.19 ± 0.73% | 93.88 ± 0.40% | 62.82 ± 2.67% | 75.62% |
| LISA | 81.09 ± 0.61% | 65.68 ± 0.87% | 94.40 ± 0.33% | 56.69 ± 1.99% | 74.47% |
| MixStyle | 83.45 ± 0.90% | 72.84 ± 0.59% | 95.20 ± 0.49% | 67.61 ± 0.83% | 79.78% |
| DDG | 79.67 ± 0.77% | 68.30 ± 0.34% | 94.72 ± 0.50% | 61.20 ± 0.82% | 75.97% |
| BODA | 81.13 ± 0.59% | 72.03 ± 0.65% | 95.73 ± 0.56% | 66.34 ± 1.55% | 78.81% |
| **TALLY (ours)** | **85.86 ± 0.40%** | **74.20 ± 0.30%** | **96.56 ± 0.20%** | **69.58 ± 0.62%** | **81.55%** |

same across different classes; (2) domain-balanced dataset, where the number of examples is the same across different domains; (3) domain-class balanced dataset, where the number of examples is the same across each domain-class group. The results are reported in Table 14, where the performance under subpopulation shift in OfficeHome-LT and DomainNet-LT and the performance under domain shift in TerraInc and iWildCam are reported. According to the results, we observe that TALLY still outperforms all variants of balanced finetuning, indicating its effectiveness in addressing multi-domain long-tailed learning problem by augmenting disentangled representations.

## F.2 Compatibility Analysis of TALLY

In this section, we analyze the compatibility of TALLY. Since TALLY only augmenting disentangled representations during the training stage, we can easily incorporate TALLY with other long-tailed learning approaches. Specifically, we incorporate TALLY with PaCo and RIDE in this analysis and report the results

Table 10: Domain shift results on OfficeHome-LT.

| | Art | Clipart | Product | Real | Avg |
|---|---|---|---|---|---|
| ERM | $45.20 \pm 0.73\%$ | $41.94 \pm 0.17\%$ | $59.21 \pm 0.44\%$ | $61.44 \pm 0.27\%$ | $51.95\%$ |
| Focal | $47.06 \pm 0.24\%$ | $43.29 \pm 0.71\%$ | $\underline{62.34 \pm 0.16\%}$ | $\underline{63.45 \pm 0.19\%}$ | $\underline{54.03\%}$ |
| LDAM | $47.08 \pm 0.37\%$ | $42.89 \pm 0.18\%$ | $61.48 \pm 0.55\%$ | $62.93 \pm 0.24\%$ | $53.60\%$ |
| CRT | $47.17 \pm 0.26\%$ | $42.62 \pm 0.55\%$ | $61.37 \pm 0.12\%$ | $63.31 \pm 0.25\%$ | $53.62\%$ |
| MiSLAS | $45.22 \pm 0.52\%$ | $41.36 \pm 0.09\%$ | $62.28 \pm 0.49\%$ | $62.56 \pm 0.25\%$ | $52.86\%$ |
| Remix | $44.26 \pm 0.49\%$ | $39.18 \pm 0.34\%$ | $60.70 \pm 0.28\%$ | $61.58 \pm 0.42\%$ | $51.43\%$ |
| RIDE | $47.15 \pm 0.38\%$ | $43.00 \pm 0.49\%$ | $61.50 \pm 0.19\%$ | $63.36 \pm 0.13\%$ | $53.75\ \%$ |
| PaCo | $\underline{48.06 \pm 0.40\%}$ | $42.61 \pm 0.83\%$ | $62.26 \pm 0.34\%$ | $62.98 \pm 0.27\%$ | $53.98\ \%$ |
| IRM | $33.55 \pm 4.21\%$ | $34.34 \pm 3.74\%$ | $49.54 \pm 5.30\%$ | $51.95 \pm 4.64\%$ | $42.34\%$ |
| GroupDRO | $44.62 \pm 0.51\%$ | $41.84 \pm 0.68\%$ | $58.40 \pm 0.43\%$ | $59.63 \pm 0.53\%$ | $51.12\%$ |
| CORAL | $43.93 \pm 0.56\%$ | $42.71 \pm 0.59\%$ | $56.91 \pm 0.45\%$ | $59.40 \pm 0.94\%$ | $50.74\%$ |
| LISA | $41.80 \pm 0.36\%$ | $36.96 \pm 0.45\%$ | $56.51 \pm 0.16\%$ | $57.62 \pm 0.39\%$ | $48.22\%$ |
| MixStyle | $45.11 \pm 0.18\%$ | $\mathbf{45.52 \pm 0.20\%}$ | $58.32 \pm 0.64\%$ | $60.92 \pm 0.22\%$ | $52.47\%$ |
| DDG | $43.89 \pm 0.39\%$ | $42.79 \pm 0.91\%$ | $57.92 \pm 0.15\%$ | $59.69 \pm 0.30\%$ | $51.07\%$ |
| BODA | $47.08 \pm 0.25\%$ | $\underline{44.38 \pm 0.77\%}$ | $59.58 \pm 0.26\%$ | $62.25 \pm 0.10\%$ | $53.32\%$ |
| **TALLY (ours)** | $\mathbf{49.79 \pm 0.76\%}$ | $44.22 \pm 0.45\%$ | $\mathbf{63.02 \pm 0.52\%}$ | $\mathbf{65.71 \pm 0.26\%}$ | $\mathbf{55.69\%}$ |

Table 11: Domain shift results on DomainNet-LT.

| | Sketch | Infograph | Painting | Quickdraw | Real | Clipart | Avg |
|---|---|---|---|---|---|---|---|
| ERM | $39.22 \pm 0.24\%$ | $18.96 \pm 0.37\%$ | $34.71 \pm 0.49\%$ | $10.70 \pm 0.06\%$ | $50.87 \pm 0.43\%$ | $44.83 \pm 0.25\%$ | $33.21\%$ |
| Focal | $41.01 \pm 0.61\%$ | $19.99 \pm 0.14\%$ | $36.42 \pm 0.45\%$ | $10.29 \pm 0.23\%$ | $\mathbf{55.63 \pm 0.56\%}$ | $48.09 \pm 0.72\%$ | $35.23\%$ |
| LDAM | $40.44 \pm 0.26\%$ | $19.06 \pm 0.20\%$ | $36.32 \pm 0.52\%$ | $11.38 \pm 0.29\%$ | $52.91 \pm 0.52\%$ | $46.38 \pm 0.44\%$ | $34.42\%$ |
| CRT | $40.78 \pm 0.35\%$ | $\underline{20.41 \pm 0.42\%}$ | $39.01 \pm 0.35\%$ | $\mathbf{11.41 \pm 0.17\%}$ | $55.26 \pm 0.69\%$ | $\underline{50.00 \pm 0.83\%}$ | $36.14\%$ |
| MiSLAS | $41.34 \pm 0.39\%$ | $19.89 \pm 0.25\%$ | $\underline{39.85 \pm 0.56\%}$ | $11.00 \pm 0.13\%$ | $\underline{55.50 \pm 0.36\%}$ | $49.49 \pm 0.29\%$ | $\underline{36.18\%}$ |
| Remix | $40.01 \pm 0.51\%$ | $19.17 \pm 0.46\%$ | $38.93 \pm 0.32\%$ | $\underline{11.39 \pm 0.20\%}$ | $53.43 \pm 0.60\%$ | $47.94 \pm 0.71\%$ | $35.14\%$ |
| RIDE | $41.12 \pm 0.57\%$ | $19.23 \pm 0.34\%$ | $37.29 \pm 0.44\%$ | $11.21 \pm 0.35\%$ | $54.00 \pm 0.63\%$ | $48.72 \pm 0.30\%$ | $35.26\%$ |
| PaCo | $40.88 \pm 0.25\%$ | $19.85 \pm 0.21\%$ | $39.11 \pm 0.18\%$ | $11.03 \pm 0.51\%$ | $55.35 \pm 0.77\%$ | $48.31 \pm 0.18\%$ | $35.76\%$ |
| IRM | $34.65 \pm 0.75\%$ | $15.41 \pm 0.83\%$ | $28.18 \pm 1.26\%$ | $7.69 \pm 0.40\%$ | $40.83 \pm 0.72\%$ | $42.36 \pm 1.25\%$ | $28.19\%$ |
| GroupDRO | $38.47 \pm 0.27\%$ | $18.63 \pm 0.07\%$ | $34.23 \pm 0.15\%$ | $10.26 \pm 0.35\%$ | $50.80 \pm 0.47\%$ | $42.85 \pm 0.63\%$ | $32.54\%$ |
| CORAL | $39.42 \pm 0.42\%$ | $19.30 \pm 0.33\%$ | $35.15 \pm 0.70\%$ | $10.61 \pm 0.22\%$ | $51.05 \pm 0.28\%$ | $45.15 \pm 0.38\%$ | $33.44\%$ |
| LISA | $40.75 \pm 0.46\%$ | $18.47 \pm 0.14\%$ | $37.99 \pm 0.19\%$ | $9.98 \pm 0.09\%$ | $54.33 \pm 0.49\%$ | $48.42 \pm 0.42\%$ | $34.99\%$ |
| MixStyle | $40.99 \pm 0.60\%$ | $18.64 \pm 0.32\%$ | $35.86 \pm 0.39\%$ | $11.03 \pm 0.15\%$ | $50.26 \pm 0.53\%$ | $45.49 \pm 0.84\%$ | $33.71\%$ |
| DDG | $40.66 \pm 0.58\%$ | $19.08 \pm 0.34\%$ | $35.61 \pm 0.63\%$ | $\underline{11.39 \pm 0.29\%}$ | $50.93 \pm 0.41\%$ | $45.95 \pm 0.47\%$ | $33.94\%$ |
| BODA | $\underline{41.95 \pm 0.45\%}$ | $\mathbf{20.65 \pm 0.58\%}$ | $37.98 \pm 0.27\%$ | $11.02 \pm 0.23\%$ | $55.22 \pm 0.65\%$ | $48.26 \pm 0.33\%$ | $35.85\%$ |
| **TALLY (ours)** | $\mathbf{42.66 \pm 0.32\%}$ | $19.26 \pm 0.09\%$ | $\mathbf{40.49 \pm 0.34\%}$ | $11.15 \pm 0.21\%$ | $54.79 \pm 0.62\%$ | $\mathbf{50.36 \pm 0.41\%}$ | $\mathbf{36.45\%}$ |

in Table 15. We observe that incorporating TALLY significantly improves the performance over the vanilla PaCo and RIDE. Nevertheless, the original TALLY already showed competitive performance compared with TALLY+PaCo and TALLY+RIDE.

### F.3   Full Results of the Effect of Prototypes

We report the full results of the prototype analysis in Table 17.

### F.4   Full Results of the Analysis of Sampling Strategies

In Table 4, we report the full results of the analysis of different sampling strategies.

Table 12: Hyperparameters for experiments on real-world data.

| Hyperparameters | TerraInc | iWildCam |
|---|---|---|
| Learning Rate | 3e-5 | 3e-5 |
| Weight Decay | 1e-6 | 0 |
| Batch Size | 18 | 16 |
| Epochs | 15 | 15 |
| Steps | 1000 | 1000 |
| Warm Start Epochs | 7 | 7 |
| $\gamma$ in feat. estimation | 0.8 | 0.8 |
| class prototype mixup parameter $\alpha_c$ | 0.5 | 0.5 |
| domain prototype mixup parameter $\alpha_d$ | 0.5 | 0.5 |

Table 13: Full Results of Domain Shifts on Real-world Data.

| | TerraInc | | iWildCam | |
|---|---|---|---|---|
| | Macro F1 | Acc | Macro F1 | Acc |
| ERM | $42.35 \pm 1.25\%$ | $54.81 \pm 0.83\%$ | $32.0 \pm 1.5\%$ | $69.0 \pm 0.4\%$ |
| Focal | $43.54 \pm 0.81\%$ | $56.62 \pm 1.49\%$ | $\underline{33.2 \pm 1.2\%}$ | $74.7 \pm 1.9\%$ |
| LDAM | $44.29 \pm 1.41\%$ | $57.22 \pm 0.92\%$ | $32.7 \pm 0.9\%$ | $\mathbf{75.2 \pm 2.0\%}$ |
| CRT | $43.09 \pm 0.79\%$ | $58.27 \pm 1.35\%$ | $32.5 \pm 1.8\%$ | $67.3 \pm 1.3\%$ |
| MiSLAS | $40.68 \pm 1.33\%$ | $52.96 \pm 2.58\%$ | $30.5 \pm 1.1\%$ | $59.8 \pm 2.8\%$ |
| Remix | $43.72 \pm 1.87\%$ | $\underline{58.40 \pm 2.57\%}$ | $28.4 \pm 0.8\%$ | $65.8 \pm 1.6\%$ |
| RIDE | $44.03 \pm 1.42\%$ | $57.89 \pm 1.46\%$ | $32.8 \pm 0.5\%$ | $70.1 \pm 1.9\%$ |
| PaCo | $43.40 \pm 1.03\%$ | $57.35 \pm 0.89\%$ | $31.9 \pm 0.2\%$ | $72.6 \pm 2.5\%$ |
| IRM | $31.17 \pm 3.52\%$ | $49.27 \pm 5.22\%$ | $15.1 \pm 4.9\%$ | $59.8 \pm 3.7\%$ |
| GroupDRO | $42.22 \pm 0.87\%$ | $56.43 \pm 1.63\%$ | $23.9 \pm 2.1\%$ | $72.7 \pm 2.0\%$ |
| CORAL | $\underline{45.43 \pm 0.92\%}$ | $58.10 \pm 1.38\%$ | $32.8 \pm 0.1\%$ | $73.3 \pm 4.3\%$ |
| LISA | $39.27 \pm 0.69\%$ | $54.92 \pm 1.04\%$ | $27.6 \pm 1.2\%$ | $64.9 \pm 2.2\%$ |
| MixStyle | $44.73 \pm 0.99\%$ | $57.55 \pm 2.05\%$ | $32.4 \pm 1.1\%$ | $\underline{74.9 \pm 2.7\%}$ |
| DDG | $40.47 \pm 1.93\%$ | $53.61 \pm 1.71\%$ | $29.8 \pm 0.2\%$ | $69.7 \pm 2.3\%$ |
| BODA | $44.47 \pm 0.84\%$ | $57.52 \pm 1.13\%$ | $32.9 \pm 0.3\%$ | $70.5 \pm 2.3\%$ |
| **TALLY (ours)** | $\mathbf{46.23 \pm 0.56\%}$ | $\mathbf{59.89 \pm 1.32\%}$ | $\mathbf{34.4 \pm 0.4\%}$ | $73.4 \pm 1.8\%$ |

# G   Results on Standard Domain Generalization Benchmarks

In this section, we present the additional comparison on standard domain generalization benchmarks. Notice that the data distributions in these standard benchmarks are not long-tailed, which is thus not our focus in this paper. The goal is to compare our approach with other domain generalization methods. In Table 19-22, we present results on four standard benchmarks: VLCS, PACS, OfficeHome, DomainNet, respectively. Results for all algorithms except TALLY are directly copied from Gulrajani & Lopez-Paz (2021) and Yang et al. (2022). In Table 23, we summarize all results and show the comparison between different approaches. According to the results, TALLY can achieve comparable performance compared with state-of-the-art domain generalization approaches.

Table 14: Performance comparison between TALLY and balanced finetuning. FT means fine-tuning. Here, worst performance means the class-balanced worst-domain accuracy.

| Subpopulation shift | OfficeHome-LT | | DomainNet-LT | |
| --- | --- | --- | --- | --- |
| | Avg. | Worst | Avg. | Worst |
| Class-balanced FT | $63.41 \pm 0.32\%$ | $58.41 \pm 0.64\%$ | $47.23 \pm 0.83\%$ | $27.65 \pm 0.05\%$ |
| Domain-balanced FT | $61.19 \pm 0.79\%$ | $54.29 \pm 0.55\%$ | $44.32 \pm 0.28\%$ | $24.37 \pm 0.13\%$ |
| Domain-class-balanced FT | $62.84 \pm 0.28\%$ | $57.88 \pm 0.82\%$ | $47.07 \pm 0.41\%$ | $27.16 \pm 0.22\%$ |
| **TALLY** | $\mathbf{67.00 \pm 0.47\%}$ | $\mathbf{60.45 \pm 0.09\%}$ | $\mathbf{50.15 \pm 0.46\%}$ | $\mathbf{29.55 \pm 0.19\%}$ |

| Domain shift | TerraInc | | iWildCam | |
| --- | --- | --- | --- | --- |
| | Macro F1 | Acc | Macro F1 | Acc |
| Class-balanced FT | $44.36 \pm 0.45\%$ | $57.85 \pm 1.91\%$ | $32.5 \pm 0.3\%$ | $72.2 \pm 2.1\%$ |
| Domain-balanced FT | $43.92 \pm 0.77\%$ | $58.12 \pm 0.73\%$ | $32.9 \pm 0.8\%$ | $\mathbf{73.7 \pm 0.7\%}$ |
| Domain-class-balanced FT | $44.21 \pm 0.31\%$ | $59.13 \pm 1.24\%$ | $32.1 \pm 0.5\%$ | $71.9 \pm 1.8\%$ |
| **TALLY** | $\mathbf{46.23 \pm 0.56\%}$ | $\mathbf{59.89 \pm 1.32\%}$ | $\mathbf{34.4 \pm 0.4\%}$ | $73.4 \pm 1.8\%$ |

Table 15: Compatibility Analysis of TALLY. Worst performance represents the class-balanced worst-domain accuracy.

| Subpopulation shift | | OfficeHome-LT | | DomainNet-LT | |
| --- | --- | --- | --- | --- | --- |
| | | Avg. | Worst | Avg. | Worst |
| Vanilla TALLY | | $67.00 \pm 0.47\%$ | $60.45 \pm 0.09\%$ | $\mathbf{50.15 \pm 0.46\%}$ | $29.55 \pm 0.19\%$ |
| RIDE | | $63.27 \pm 0.54\%$ | $54.04 \pm 0.28\%$ | $47.51 \pm 0.83\%$ | $28.16 \pm 0.60\%$ |
| | +TALLY | $66.20 \pm 0.25\%$ | $60.38 \pm 0.37\%$ | $49.82 \pm 0.78\%$ | $28.72 \pm 0.10\%$ |
| PaCo | | $63.03 \pm 0.92\%$ | $56.15 \pm 0.71\%$ | $48.26 \pm 0.22\%$ | $27.36 \pm 0.52\%$ |
| | +TALLY | $\mathbf{67.30 \pm 0.22\%}$ | $\mathbf{60.62 \pm 0.57\%}$ | $50.01 \pm 0.19\%$ | $\mathbf{29.70 \pm 0.29\%}$ |

| Domain shift | | TerraInc | | iWildCam | |
| --- | --- | --- | --- | --- | --- |
| | | Macro F1 | Acc | Macro F1 | Acc |
| Vanilla TALLY | | $46.23 \pm 0.56\%$ | $\mathbf{59.89 \pm 1.32\%}$ | $\mathbf{34.4 \pm 0.4\%}$ | $73.4 \pm 1.8\%$ |
| RIDE | | $44.03 \pm 1.42\%$ | $57.89 \pm 1.46\%$ | $32.8 \pm 0.5\%$ | $70.1 \pm 1.9\%$ |
| | +TALLY | $45.76 \pm 0.38\%$ | $59.41 \pm 1.24\%$ | $33.4 \pm 0.3\%$ | $\mathbf{73.7 \pm 1.4\%}$ |
| PaCo | | $43.40 \pm 1.03\%$ | $57.35 \pm 0.89\%$ | $31.9 \pm 0.2\%$ | $72.6 \pm 2.5\%$ |
| | +TALLY | $\mathbf{46.46 \pm 0.21\%}$ | $59.19 \pm 0.82\%$ | $33.9 \pm 0.7\%$ | $72.6 \pm 2.1\%$ |

Table 16: Full results of the comparison between TALLY and variants of two representative domain generalization approaches (CORAL, MixStyle). Worst means class-balanced worst domain accuracy in subpopulation shift.

| Subpopulation shift | | OfficeHome-LT | | DomainNet-LT | |
|---|---|---|---|---|---|
| | | Avg. | Worst | Avg. | Worst |
| CORAL | | $59.10 \pm 0.20\%$ | $53.53 \pm 0.60\%$ | $43.92 \pm 0.36\%$ | $24.50 \pm 0.68\%$ |
| | +UW | $60.91 \pm 1.40\%$ | $55.32 \pm 2.14\%$ | $46.26 \pm 0.48\%$ | $26.74 \pm 0.15\%$ |
| | +Focal | $61.72 \pm 0.93\%$ | $55.26 \pm 1.60\%$ | $46.39 \pm 0.31\%$ | $27.13 \pm 0.95\%$ |
| | +LDAM | $61.09 \pm 1.52\%$ | $56.22 \pm 1.45\%$ | $46.69 \pm 0.32\%$ | $26.58 \pm 0.62\%$ |
| | +CRT | $62.61 \pm 0.68\%$ | $56.41 \pm 0.51\%$ | $47.68 \pm 0.53\%$ | $27.60 \pm 0.37\%$ |
| MixStyle | | $62.26 \pm 0.22\%$ | $55.19 \pm 1.10\%$ | $43.59 \pm 0.57\%$ | $22.65 \pm 0.22\%$ |
| | +UW | $63.02 \pm 0.69\%$ | $57.50 \pm 2.11\%$ | $46.30 \pm 0.35\%$ | $25.53 \pm 0.48\%$ |
| | +Focal | $63.69 \pm 0.32\%$ | $57.56 \pm 1.40\%$ | $46.18 \pm 0.36\%$ | $26.51 \pm 0.88\%$ |
| | +LDAM | $62.77 \pm 0.78\%$ | $57.50 \pm 1.50\%$ | $46.32 \pm 0.35\%$ | $25.31 \pm 0.48\%$ |
| | +CRT | $63.20 \pm 0.29\%$ | $55.77 \pm 0.63\%$ | $48.11 \pm 0.29\%$ | $27.95 \pm 0.72\%$ |
| **TALLY** | | $\mathbf{67.00 \pm 0.47\%}$ | $\mathbf{60.45 \pm 0.09\%}$ | $\mathbf{50.15 \pm 0.46\%}$ | $\mathbf{29.55 \pm 0.19\%}$ |

| Domain shift | | TerraInc | | iWildCam | |
|---|---|---|---|---|---|
| | | Macro F1 | Acc | Macro F1 | Acc |
| CORAL | | $45.43 \pm 0.92\%$ | $58.10 \pm 1.38\%$ | $32.8 \pm 0.1\%$ | $73.3 \pm 4.3\%$ |
| | +UW | $45.67 \pm 0.73\%$ | $59.73 \pm 1.53\%$ | $32.6 \pm 0.7\%$ | $73.0 \pm 2.9\%$ |
| | +Focal | $45.27 \pm 0.88\%$ | $59.54 \pm 1.71\%$ | $32.9 \pm 0.8\%$ | $74.9 \pm 3.2\%$ |
| | +LDAM | $45.45 \pm 1.13\%$ | $59.50 \pm 2.04\%$ | $33.4 \pm 0.5\%$ | $74.3 \pm 1.6\%$ |
| | +CRT | $45.72 \pm 1.59\%$ | $59.47 \pm 2.32\%$ | $33.2 \pm 0.6\%$ | $73.2 \pm 5.3\%$ |
| MixStyle | | $44.73 \pm 0.99\%$ | $57.55 \pm 2.05\%$ | $32.4 \pm 1.1\%$ | $74.9 \pm 2.7\%$ |
| | +UW | $45.25 \pm 0.67\%$ | $58.65 \pm 1.39\%$ | $32.7 \pm 0.9\%$ | $72.0 \pm 1.3\%$ |
| | +Focal | $45.04 \pm 1.22\%$ | $58.15 \pm 2.18\%$ | $32.8 \pm 0.5\%$ | $74.0 \pm 2.8\%$ |
| | +LDAM | $44.74 \pm 1.03\%$ | $58.66 \pm 1.76\%$ | $32.7 \pm 1.3\%$ | $\mathbf{77.1 \pm 3.0\%}$ |
| | +CRT | $44.97 \pm 1.83\%$ | $58.38 \pm 3.11\%$ | $33.1 \pm 0.7\%$ | $71.7 \pm 3.3\%$ |
| **TALLY** | | $\mathbf{46.23 \pm 0.56\%}$ | $\mathbf{59.89 \pm 1.32\%}$ | $\mathbf{34.4 \pm 0.4\%}$ | $73.4 \pm 1.8\%$ |

Table 17: Full results of the analysis of prototype-guided invariant learning. C Only and D Only represent only using class prototype representation or class-agnostic domain factors, respectively. Worst means class-balanced worst domain accuracy in subpopulation shift.

| Subpopulation shift | OfficeHome-LT | | DomainNet-LT | |
|---|---|---|---|---|
| | Avg. | Worst | Avg. | Worst |
| None | $66.19 \pm 0.34\%$ | $58.72 \pm 0.89\%$ | $48.78 \pm 0.43\%$ | $27.21 \pm 0.12\%$ |
| C Only | $66.54 \pm 0.14\%$ | $59.55 \pm 0.55\%$ | $49.45 \pm 0.27\%$ | $27.50 \pm 0.38\%$ |
| D Only | $66.23 \pm 0.24\%$ | $58.98 \pm 1.18\%$ | $49.09 \pm 0.54\%$ | $27.50 \pm 0.46\%$ |
| **TALLY** | $\mathbf{67.00 \pm 0.47\%}$ | $\mathbf{60.45 \pm 0.09\%}$ | $\mathbf{50.15 \pm 0.46\%}$ | $\mathbf{29.55 \pm 0.19\%}$ |

| Domain shift | TerraInc | | iWildCam | |
|---|---|---|---|---|
| | Macro F1 | Acc | Macro F1 | Acc |
| None | $45.30 \pm 0.64\%$ | $58.02 \pm 1.26\%$ | $32.9 \pm 0.8\%$ | $72.6 \pm 2.9\%$ |
| C Only | $45.86 \pm 0.81\%$ | $59.06 \pm 1.46\%$ | $33.9 \pm 0.5\%$ | $\mathbf{74.6 \pm 2.8\%}$ |
| D Only | $45.47 \pm 0.57\%$ | $58.22 \pm 1.84\%$ | $33.2 \pm 1.1\%$ | $72.3 \pm 1.5\%$ |
| **TALLY** | $\mathbf{46.23 \pm 0.56\%}$ | $\mathbf{59.89 \pm 1.32\%}$ | $\mathbf{34.4 \pm 0.4\%}$ | $73.4 \pm 1.8\%$ |

Table 18: Full results of comparison between sampling strategies. Worst means class-balanced worst domain accuracy in subpopulation shift.

| Subpopulation shift | OfficeHome-LT | | DomainNet-LT | |
|---|---|---|---|---|
| | Avg. | Worst | Avg. | Worst |
| Balanced Sampling | $65.03 \pm 0.91\%$ | $58.33 \pm 0.24\%$ | $49.35 \pm 0.21\%$ | $27.97 \pm 0.25\%$ |
| **TALLY**(Selective)) | $\mathbf{67.00 \pm 0.47\%}$ | $\mathbf{60.45 \pm 0.09\%}$ | $\mathbf{50.15 \pm 0.46\%}$ | $\mathbf{29.55 \pm 0.19\%}$ |

| Domain shift | TerraInc | | iWildCam | |
|---|---|---|---|---|
| | Macro F1 | Acc | Macro F1 | Acc |
| Balanced Sampling | $44.79 \pm 0.62\%$ | $57.78 \pm 0.36\%$ | $33.1 \pm 0.4\%$ | $71.6 \pm 1.8\%$ |
| **TALLY**(Selective)) | $\mathbf{46.23 \pm 0.56\%}$ | $\mathbf{59.89 \pm 1.32\%}$ | $\mathbf{34.4 \pm 0.4\%}$ | $73.4 \pm 1.8\%$ |

Table 19: Comparison on the standard VLCS benchmark.

| | Caltech101 | LabelMe | SUN09 | VOC2007 | Avg |
|---|---|---|---|---|---|
| ERM | $97.7 \pm 0.4$ | $64.3 \pm 0.9$ | $73.4 \pm 0.5$ | $74.6 \pm 1.3$ | 77.5 |
| IRM | $98.6 \pm 0.1$ | $64.9 \pm 0.9$ | $73.4 \pm 0.6$ | $77.3 \pm 0.9$ | 78.5 |
| GroupDRO | $97.3 \pm 0.3$ | $63.4 \pm 0.9$ | $69.5 \pm 0.8$ | $76.7 \pm 0.7$ | 76.7 |
| Mixup | $98.3 \pm 0.6$ | $64.8 \pm 1.0$ | $72.1 \pm 0.5$ | $74.3 \pm 0.8$ | 77.4 |
| MLDG | $97.4 \pm 0.2$ | $65.2 \pm 0.7$ | $71.0 \pm 1.4$ | $75.3 \pm 1.0$ | 77.2 |
| CORAL | $98.3 \pm 0.1$ | $\underline{66.1 \pm 1.2}$ | $73.4 \pm 0.3$ | $77.5 \pm 1.2$ | **78.8** |
| MMD | $97.7 \pm 0.1$ | $64.0 \pm 1.1$ | $72.8 \pm 0.2$ | $75.3 \pm 3.3$ | 77.5 |
| DANN | $\mathbf{99.0 \pm 0.3}$ | $65.1 \pm 1.4$ | $73.1 \pm 0.3$ | $77.2 \pm 0.6$ | $\underline{78.6}$ |
| CDANN | $97.1 \pm 0.3$ | $65.1 \pm 1.2$ | $70.7 \pm 0.8$ | $77.1 \pm 1.5$ | 77.5 |
| MTL | $97.8 \pm 0.4$ | $64.3 \pm 0.3$ | $71.5 \pm 0.7$ | $75.3 \pm 1.7$ | 77.2 |
| SagNet | $97.9 \pm 0.4$ | $64.5 \pm 0.5$ | $71.4 \pm 1.3$ | $77.5 \pm 0.5$ | 77.8 |
| ARM | $\underline{98.7 \pm 0.2}$ | $63.6 \pm 0.7$ | $71.3 \pm 1.2$ | $76.7 \pm 0.6$ | 77.6 |
| VREx | $98.4 \pm 0.3$ | $64.4 \pm 1.4$ | $\underline{74.1 \pm 0.4}$ | $76.2 \pm 1.3$ | 78.3 |
| RSC | $97.9 \pm 0.1$ | $62.5 \pm 0.7$ | $\underline{72.3 \pm 1.2}$ | $75.6 \pm 0.8$ | 77.1 |
| BODA | $98.1 \pm 0.3$ | $64.5 \pm 0.4$ | $\mathbf{74.3 \pm 0.3}$ | $\underline{78.0 \pm 0.6}$ | 78.5 |
| **TALLY (ours)** | $97.5 \pm 0.5$ | $\mathbf{67.2 \pm 1.1}$ | $73.8 \pm 0.5$ | $\mathbf{79.2 \pm 0.9}$ | **78.8** |

Table 20: Comparison on the standard PACS benchmark.

| | Art painting | Cartoon | Photo | Sketch | Avg |
|---|---|---|---|---|---|
| ERM | $84.7 \pm 0.4$ | $80.8 \pm 0.6$ | $97.2 \pm 0.3$ | $79.3 \pm 1.0$ | 85.5 |
| IRM | $84.8 \pm 1.3$ | $76.4 \pm 1.1$ | $96.7 \pm 0.6$ | $76.1 \pm 1.0$ | 83.5 |
| GroupDRO | $83.5 \pm 0.9$ | $79.1 \pm 0.6$ | $96.7 \pm 0.3$ | $78.3 \pm 2.0$ | 84.4 |
| Mixup | $86.1 \pm 0.5$ | $78.9 \pm 0.8$ | $\underline{97.6 \pm 0.1}$ | $75.8 \pm 1.8$ | 84.6 |
| MLDG | $85.5 \pm 1.4$ | $80.1 \pm 1.7$ | $97.4 \pm 0.3$ | $76.6 \pm 1.1$ | 84.9 |
| CORAL | $\underline{88.3 \pm 0.2}$ | $80.0 \pm 0.5$ | $97.5 \pm 0.3$ | $78.8 \pm 1.3$ | 86.2 |
| MMD | $86.1 \pm 1.4$ | $79.4 \pm 0.9$ | $96.6 \pm 0.2$ | $76.5 \pm 0.5$ | 84.6 |
| DANN | $86.4 \pm 0.8$ | $77.4 \pm 0.8$ | $97.3 \pm 0.4$ | $73.5 \pm 2.3$ | 83.6 |
| CDANN | $84.6 \pm 1.8$ | $75.5 \pm 0.9$ | $96.8 \pm 0.3$ | $73.5 \pm 0.6$ | 82.6 |
| MTL | $87.5 \pm 0.8$ | $77.1 \pm 0.5$ | $96.4 \pm 0.8$ | $77.3 \pm 1.8$ | 84.6 |
| SagNet | $87.4 \pm 1.0$ | $80.7 \pm 0.6$ | $97.1 \pm 0.1$ | $80.0 \pm 0.4$ | 86.3 |
| ARM | $86.8 \pm 0.6$ | $76.8 \pm 0.5$ | $97.4 \pm 0.3$ | $79.3 \pm 1.2$ | 85.1 |
| VREx | $86.0 \pm 1.6$ | $79.1 \pm 0.6$ | $96.9 \pm 0.5$ | $77.7 \pm 1.7$ | 84.9 |
| RSC | $85.4 \pm 0.8$ | $79.7 \pm 1.8$ | $\underline{97.6 \pm 0.3}$ | $78.2 \pm 1.2$ | 85.2 |
| BODA | $88.2 \pm 0.2$ | $\mathbf{81.7 \pm 0.3}$ | $\mathbf{97.8 \pm 0.2}$ | $\underline{80.2 \pm 0.3}$ | $\underline{86.9}$ |
| **TALLY (ours)** | $\mathbf{89.5 \pm 0.8}$ | $\underline{81.2 \pm 0.7}$ | $97.0 \pm 0.1$ | $\mathbf{81.7 \pm 0.9}$ | **87.4** |

Table 21: Comparison on the standard OfficeHome benchmark.

|  | Art | Clipart | Product | Real | Avg |
|---|---|---|---|---|---|
| ERM | 61.3 ± 0.7 | 52.4 ± 0.3 | 75.8 ± 0.1 | 76.6 ± 0.3 | 66.5 |
| IRM | 58.9 ± 2.3 | 52.2 ± 1.6 | 72.1 ± 2.9 | 74.0 ± 2.5 | 64.3 |
| GroupDRO | 60.4 ± 0.7 | 52.7 ± 1.0 | 75.0 ± 0.7 | 76.0 ± 0.7 | 66.0 |
| Mixup | 62.4 ± 0.8 | 54.8 ± 0.6 | 76.9 ± 0.3 | 78.3 ± 0.2 | 68.1 |
| MLDG | 61.5 ± 0.9 | 53.2 ± 0.6 | 75.0 ± 1.2 | 77.5 ± 0.4 | 66.8 |
| CORAL | 65.3 ± 0.4 | 54.4 ± 0.5 | 76.5 ± 0.1 | 78.4 ± 0.5 | 68.7 |
| MMD | 60.4 ± 0.2 | 53.3 ± 0.3 | 74.3 ± 0.1 | 77.4 ± 0.6 | 66.3 |
| DANN | 59.9 ± 1.3 | 53.0 ± 0.3 | 73.6 ± 0.7 | 76.9 ± 0.5 | 65.9 |
| CDANN | 61.5 ± 1.4 | 50.4 ± 2.4 | 74.4 ± 0.9 | 76.6 ± 0.8 | 65.8 |
| MTL | 61.5 ± 0.7 | 52.4 ± 0.6 | 74.9 ± 0.4 | 76.8 ± 0.4 | 66.4 |
| SagNet | 63.4 ± 0.2 | 54.8 ± 0.4 | 75.8 ± 0.4 | 78.3 ± 0.3 | 68.1 |
| ARM | 58.9 ± 0.8 | 51.0 ± 0.5 | 74.1 ± 0.1 | 75.2 ± 0.3 | 64.8 |
| VREx | 60.7 ± 0.9 | 53.0 ± 0.9 | 75.3 ± 0.1 | 76.6 ± 0.5 | 66.4 |
| RSC | 60.7 ± 1.4 | 51.4 ± 0.3 | 74.8 ± 1.1 | 75.1 ± 1.3 | 65.5 |
| BODA | **65.4 ± 0.1** | **55.4 ± 0.3** | 77.1 ± 0.1 | **79.5 ± 0.3** | **69.3** |
| **TALLY (ours)** | 64.2 ± 0.5 | 55.1 ± 0.8 | **78.0 ± 1.1** | 79.2 ± 0.5 | 69.1 |

Table 22: Comparison on the standard DomainNet benchmark.

|  | Sketch | Infograph | Painting | Quickdraw | Real | Clipart | Avg |
|---|---|---|---|---|---|---|---|
| ERM | 49.8 ± 0.4 | 18.8 ± 0.3 | 46.7 ± 0.3 | 12.2 ± 0.4 | 59.6 ± 0.1 | 58.1 ± 0.3 | 40.9 |
| IRM | 42.3 ± 3.1 | 15.0 ± 1.5 | 38.3 ± 4.3 | 10.9 ± 0.5 | 48.2 ± 5.2 | 48.5 ± 2.8 | 33.9 |
| GroupDRO | 40.1 ± 0.6 | 17.5 ± 0.4 | 33.8 ± 0.5 | 9.3 ± 0.3 | 51.6 ± 0.4 | 47.2 ± 0.5 | 33.3 |
| Mixup | 48.2 ± 0.5 | 18.5 ± 0.5 | 44.3 ± 0.5 | 12.5 ± 0.4 | 55.8 ± 0.3 | 55.7 ± 0.3 | 39.2 |
| MLDG | 50.2 ± 0.4 | 19.1 ± 0.3 | 45.8 ± 0.7 | 13.4 ± 0.3 | 59.6 ± 0.2 | 59.1 ± 0.2 | 41.2 |
| CORAL | 50.1 ± 0.6 | 19.7 ± 0.2 | 46.6 ± 0.3 | 13.4 ± 0.4 | 59.8 ± 0.2 | 59.2 ± 0.1 | 41.5 |
| MMD | 28.9 ± 11.9 | 11.0 ± 4.6 | 26.8 ± 11.3 | 8.7 ± 2.1 | 32.7 ± 13.8 | 32.1 ± 13.3 | 23.4 |
| DANN | 46.8 ± 0.6 | 18.3 ± 0.1 | 44.2 ± 0.7 | 11.8 ± 0.1 | 55.5 ± 0.4 | 53.1 ± 0.2 | 38.3 |
| CDANN | 45.9 ± 0.5 | 17.3 ± 0.1 | 43.7 ± 0.9 | 12.1 ± 0.7 | 56.2 ± 0.4 | 54.6 ± 0.4 | 38.3 |
| MTL | 49.2 ± 0.1 | 18.5 ± 0.4 | 46.0 ± 0.1 | 12.5 ± 0.1 | 59.5 ± 0.3 | 57.9 ± 0.5 | 40.6 |
| SagNet | 48.8 ± 0.2 | 19.0 ± 0.2 | 45.3 ± 0.3 | 12.7 ± 0.5 | 58.1 ± 0.5 | 57.7 ± 0.3 | 40.3 |
| ARM | 43.5 ± 0.4 | 16.3 ± 0.5 | 40.9 ± 1.1 | 9.4 ± 0.1 | 53.4 ± 0.4 | 49.7 ± 0.3 | 35.5 |
| VREx | 42.0 ± 3.0 | 16.0 ± 1.5 | 35.8 ± 4.6 | 10.9 ± 0.3 | 49.6 ± 4.9 | 47.3 ± 3.5 | 33.6 |
| RSC | 47.8 ± 0.9 | 18.3 ± 0.5 | 44.4 ± 0.6 | 12.2 ± 0.2 | 55.7 ± 0.7 | 55.0 ± 1.2 | 38.9 |
| BODA | **51.3 ± 0.3** | **20.5 ± 0.7** | **48.0 ± 0.1** | 13.8 ± 0.6 | **60.6 ± 0.4** | **62.1 ± 0.4** | **42.7** |
| **TALLY (ours)** | 50.5 ± 0.2 | 19.7 ± 0.1 | 47.7 ± 0.6 | **14.1 ± 0.3** | 60.0 ± 0.2 | 60.1 ± 0.5 | 42.0 |

Table 23: Domain shift results over all four benchmarks.

|  | VLCS | PACS | OfficeHome | DomainNet | Avg |
|---|---|---|---|---|---|
| ERM | 77.5 ± 0.4 | 85.5 ± 0.2 | 66.5 ± 0.3 | 40.9 ± 0.1 | 67.6 |
| IRM | 78.5 ± 0.5 | 83.5 ± 0.8 | 64.3 ± 2.2 | 33.9 ± 2.8 | 65.1 |
| GroupDRO | 76.7 ± 0.6 | 84.4 ± 0.8 | 66.0 ± 0.7 | 33.3 ± 0.2 | 65.1 |
| Mixup | 77.4 ± 0.6 | 84.6 ± 0.6 | 68.1 ± 0.3 | 39.2 ± 0.1 | 67.3 |
| MLDG | 77.2 ± 0.4 | 84.9 ± 1.0 | 66.8 ± 0.6 | 41.2 ± 0.1 | 67.5 |
| CORAL | **78.8 ± 0.6** | 86.2 ± 0.3 | 68.7 ± 0.3 | 41.5 ± 0.1 | 68.8 |
| MMD | 77.5 ± 0.9 | 84.6 ± 0.5 | 66.3 ± 0.1 | 23.4 ± 9.5 | 63.0 |
| DANN | 78.6 ± 0.4 | 83.6 ± 0.4 | 65.9 ± 0.6 | 38.3 ± 0.1 | 66.6 |
| CDANN | 77.5 ± 0.1 | 82.6 ± 0.9 | 65.8 ± 1.3 | 38.3 ± 0.3 | 66.3 |
| MTL | 77.2 ± 0.4 | 84.6 ± 0.5 | 66.4 ± 0.5 | 40.6 ± 0.1 | 67.2 |
| SagNet | 77.8 ± 0.5 | 86.3 ± 0.2 | 68.1 ± 0.1 | 40.3 ± 0.1 | 68.1 |
| ARM | 77.6 ± 0.3 | 85.1 ± 0.4 | 64.8 ± 0.3 | 35.5 ± 0.2 | 65.8 |
| VREx | 78.3 ± 0.2 | 84.9 ± 0.6 | 66.4 ± 0.6 | 33.6 ± 2.9 | 65.8 |
| RSC | 77.1 ± 0.5 | 85.2 ± 0.9 | 65.5 ± 0.9 | 38.9 ± 0.5 | 66.7 |
| BODA | 78.5 ± 0.3 | 86.9 ± 0.4 | **69.3 ± 0.1** | **42.7 ± 0.1** | **69.4** |
| **TALLY (ours)** | **78.8 ± 0.4** | **87.4 ± 0.2** | 69.1 ± 0.4 | 42.0 ± 0.1 | 69.3 |

