# OpenReview forum: "Multi-Domain Long-Tailed Learning by Augmenting Disentangled Representations"
_TMLR — Accepted by TMLR_

### Review · Reviewer_pBD7 · 2023-07-08

**Summary Of Contributions:**

In real-world classification problems, there’s a pervasive issue known as class imbalance, which becomes even more complex when considering multiple domains, each with its own class imbalance. To address this, the authors introduce TALLY, a new method to deal with the multi-domain long-tailed learning problem. TALLY employs a selective balanced sampling strategy to mix the semantic representation of one example with the domain-related nuisances of another, creating a new representation for data augmentation. Additionally, TALLY uses a domain-invariant class prototype that eliminates domain-specific effects. Upon evaluation of multiple benchmarks and real-world datasets, TALLY consistently surpasses other baselines in handling subpopulations and domain shifts.

**Audience:**

Yes

**Broader Impact Concerns:**

I do not have concerns about this paper. I believe that developing machine learning models capable of effectively addressing challenges such as subpopulation-shift, multiple domains, and nuisances is important for the field. These advancements will significantly contribute to the development of more robust and safer models, benefiting various domains and applications.

**Claims And Evidence:**

Yes

**Requested Changes:**

The paper is well-written, and the extensive validation in the experimental section provides strong support for the method. Therefore, I would accept the paper in its current form, and I have no critical recommendations for acceptance.

**Notes**:

* Regarding the use of disentanglement, it is important to note that the term "disentanglement" has slightly different interpretations in the machine learning field depending on the context. In this case, the method appears to focus more on debiasing and domain separation, as depicted in Equation 2 and Equation 3. Other studies in the literature have proposed similar class/nuisance separation methods without explicitly using the disentanglement notation (like LISA or GroupDRO). Moreover, the paper does not focus on methods that explicitly perform disentanglement, such as those found in the object-centric literature (see [here](https://proceedings.neurips.cc/paper_files/paper/2019/hash/bc3c4a6331a8a9950945a1aa8c95ab8a-Abstract.html) and [here](https://arxiv.org/abs/2010.14407)). Introducing concepts from the disentanglement literature seems unnecessarily complex and does not provide additional value to the paper.

* In Appendix B, the limitations of mixing mechanisms like LISA are discussed. While there is some merit to your point, it is not entirely convincing that the limitations are primarily related to the mixing procedure itself. Rather, the limitations may be more closely tied to the supervised selection rule applied to choose the samples for augmentation. If, instead of using full supervision, samples were selected for augmentation based on a self-supervised criterion (such as [loss magnitude](https://proceedings.mlr.press/v139/liu21f.html)) during the early stages of training, the augmented samples would likely exhibit lower correlation with classes and domains while demonstrating a greater correlation with challenging or easy-to-learn instances.


**Questions**:

* How sensitive is the method to the quality of the class prototype? Have you experimented with using different models for the prototypes?

* If I understand correctly, the method leverages fine-grained class and domain information during augmentation. Have you explored experiments with self-supervised and weakly-supervised approaches, such as [JTT](https://proceedings.mlr.press/v139/liu21f.html) and [JM1](https://openreview.net/forum?id=HI2ilxFli0W)?

* Following up on the previous question, I am curious about the consistently good performance of the ERM estimator in the long-tail scenario across all the benchmark datasets in both the main paper and the appendix. In my experience, when dealing with subpopulation-shift and worst-group generalization, the ERM estimator does not perform particularly well. Do you have any intuition or explanation for this strong performance? My intuition suggests that the performance of the ERM estimator improves as the training distribution becomes more diverse in terms of classes and domains, and that many benchmark datasets may be too easy or artificially designed to properly evaluate the generalization capability of the ERM estimator.

**Strengths And Weaknesses:**

**Strengths**:

* The method is presented in a simple and clear manner, making it easy to understand. The concept of using linear mixing to augment dataset representations and address domain-shift and imbalance is particularly appealing.

* The notation used in the paper is clear and the formulation of the method is well-presented. This helps readers grasp the ideas and algorithms described in the paper more effectively.

* The experimental section, both in the main paper and the appendix, is extensive and thorough. The results clearly demonstrate the advantages of leveraging TALLY for long-tail multi-domain problems. Additionally, the method is compared against strong baselines, which adds credibility to its effectiveness.

**Weaknesses**:

* The novelty of the method is somewhat limited. It combines ideas from LISA and prototypes in few-shot learning literature. However, despite this limitation, the extensive experimental results and ablations presented in the paper make a significant contribution to the research community.

---

### Review · Reviewer_aSKC · 2023-07-29

**Summary Of Contributions:**

The paper presents a new method called TALLY for tackling the problem of imbalanced class distributions across different domains. TALLY incorporates a selective balanced sampling strategy and prototype-guided invariant learning. The paper shows that TALLY consistently outperforms other methods in improving robustness to subpopulation shifts. It is particularly effective for class-imbalanced problems, with larger improvements on smaller classes rather than across the board.

**Audience:**

Yes

**Broader Impact Concerns:**

NA.

**Claims And Evidence:**

Yes

**Requested Changes:**

Clarify the role of balanced augmentation as a form of regularization: The paper suggests that designing regularizers for diverse datasets can be challenging. However, it would be helpful to clarify the role of balanced augmentation as a form of regularization, as this could address some of the challenges mentioned. This adjustment is not critical for acceptance but would strengthen the work.

**Strengths And Weaknesses:**


Strengths:
- TALLY consistently outperforms other methods in improving robustness to subpopulation shifts.
- The method is particularly effective for class-imbalanced problems, with larger improvements on smaller classes.
- TALLY shows superior performance compared to other methods in dealing with multi-domain long-tailed learning.
- The paper demonstrates that TALLY outperforms its variants, verifying the effectiveness of prototype representation in mitigating nuisances.

Weaknesses:
- While TALLY outperforms other methods, the improvements are more significant for smaller classes, suggesting that the method may not improve performance across the board.
- The paper suggests that designing regularizers suitable for datasets from diverse domains can be challenging. However, I thought that balanced augmentation itself can be regarded as one kind of regularization.
- The complexity of the TALLY method might be a potential weakness. The implementation and computation might be more complex compared to other simpler methods, which could be a barrier for some applications.

---

### Review · Reviewer_TxEH · 2023-08-24

**Summary Of Contributions:**

In this work, authors attempt to tackle an important but under-explored research area in few-shot learning called "Multi-Domain Long-Tailed Learning". The proposed problem statement attempts to unify the two existing research topics -- Long-tailed Learning and Multi-Domain Learning both of which comes with its own challenges. As a solution to this problem, authors propose an algorithm called TALLY which is a mixture of three components -- representation modification to combine and split domain-specific and domain-agnostic information, a modified balanced sampling scheme and further augmentation the representation by class prototypes. For the experiments part, as the proposed task is somewhat novel and no direct benchmark exists, authors modify existing long-tailed datasets to create a multi-domain long-tailed dataset and also attempt to retrofit existing algorithms to this newly created benchmark.  Experimental results on this newly created benchmark shows TALLY outperforms existing methods when applied to this benchmark.

**Audience:**

Yes

**Broader Impact Concerns:**

N/A.

**Claims And Evidence:**

Yes

**Requested Changes:**

Please see my comment around weaknesses -- my main suggestion is to make the intuition/reasoning more clear regarding various components of the TALLY algorithm.

**Strengths And Weaknesses:**

Strengths
------------
* The main strength for this paper is the problem formulation -- I think it is novel and practical and is closer to real-world use-cases. Having an efficient algorithm that can solve such a task well will be in general useful. At the same time, the proposed experimental setup is hard and existing algorithms (including TALLY) are quite far from solving the tasks - therefore, creating future research opportunities as well.
* The proposed algorithm is easy to understand, well explained and does not seem to have any difficulty in being adopted by practitioners.
* Experimental results are robust - sufficient amount of ablations have been performed for different components of the algorithm and overall  confirms the efficacy of the method.

Weaknesses
---------------
*  The main weakness in my opinion is that the algorithm TALLY is not well motivated. The claim that instance-norm helps to disentangle domain-specific and nuisance attributes is not intuitively clear to me. No specific experiments have been performed to provide more intuition behind this which is the foundation of the algorithm.
* Along the same line, the third component (we can call it prototype-based rectification) is again not well-motivated. Although it is ablated via an experiment, not enough justification provided regarding the cause -- If nuisance attributes can be separated out by the first component, why do we still need another component.
* The idea of interpolation based sample augmentation has been proposed in MixUp and a series of subsequent works - that component of the algorithm is not novel. Although I must mention that all the related works have been clearly mentioned in the paper.

---

### Author Response · Authors · 2023-10-06
**Code Link**

The code and data is released at: https://github.com/huaxiuyao/TALLY

---

### Decision · Action_Editors · 2023-09-30

**Recommendation:** Accept as is

**Comment:**

This paper presents a new method called TALLY to deal with the multi-domain long-tailed learning problem. After rebuttal, it received Accept, Leaning Accept, and Leaning Reject recommendations. On one hand, reviewers commented that (1) this paper tackles an important problem, i.e., class imbalance in the presence of multiple domains. (2) The method is presented in a clear manner, and the paper is well written. (3) The proposed method is well justified with strong empirical results, and the contribution is relevant to the community. On the other hand, some reviewers also showed concerns regarding the novelty of the overall method as well as the intuition behind some of the components which were not clarified enough by the authors in their rebuttal.

Overall, the action editor thinks that the authors have done a good job of rebuttal, and the merits of this paper outweigh its flaws, therefore, would like to recommend acceptance of the paper by the end.

**Audience:**

Yes, researchers working in this specific sub-field will be interested in this paper.

**Claims And Evidence:**

The claims made in the submission are supported by convincing evidence.